# DIVID: DISENTANGLED SPATIO-TEMPORAL MODELING WITHIN LLMS FOR TEMPORALLY GROUNDED VIDEO UNDERSTANDING

**Yepeng Tang**[1,2*] **, Weining Wang**[3,4]**, Longteng Guo**[3,4]**, Tongtian Yue**[3,4]**, Wenxuan Wang**[3,4]**,
Chunjie Zhang**[1,2†]**, Jing Liu**[3,4†]

[1]Institute of Information Science, School of Computer Science and Technology,
Beijing Jiaotong University
[2]Visual Intelligence + X International Cooperation Joint Laboratory of MOE,
School of Computer Science and Technology, Beijing Jiaotong University
[3]Institute of Automation, Chinese Academy of Sciences
[4]School of Artificial Intelligence, University of Chinese Academy of Sciences

## ABSTRACT

Recent advances in Video LLMs have improved video understanding performance, but temporally grounded understanding in long-form videos remains challenging. Most models encode video frames into a flat sequence of visual tokens, which are then processed together with textual input by the LLM. While effective for short videos, this approach becomes inefficient for long-form videos due to lengthy token sequences that exceed context limits and incur high computational costs. Slow-Fast architectures partially address this by separating temporal and spatial features during encoding, but these features are still processed jointly within the LLM, lacking true spatio-temporal disentanglement. Moreover, spatial features are typically sampled in a query-agnostic manner, risking the loss of task-relevant content. To address these limitations, we propose Divid, a novel dual-branch framework that explicitly disentangles spatial and temporal modeling within the LLM decoder. Specifically, the temporal branch processes densely sampled, low-resolution frames to effectively capture long-range motion dynamics, while the spatial branch selects a sparse set of high-resolution keyframes guided by temporal attention. To unify the two branches, we design a lightweight spatio-temporal soft-router that adaptively fuses temporal and spatial cues at the token level, conditioned on the input query. This disentangled architecture not only improves temporal alignment accuracy but also leads to computational savings by minimizing redundant visual processing. Furthermore, we introduce TempGCap, a large-scale dataset consisting of 559K timestamp-grounded video-text pairs, providing rich temporal supervision. Extensive experiments on temporal grounding and grounded videoQA benchmarks demonstrate the superior performance and efficiency of our proposed Divid.

## 1 INTRODUCTION

Recent advances in video-language models with large language models (Video LLMs) have significantly improved the performance of video understanding tasks (Li et al., 2025b; Liu et al., 2024a; 2025). Despite these advances, temporally grounded video understanding in long-form videos remains a challenging problem. These tasks require models to reason about complex spatio-temporal structures and align visual content with language instructions accurately at temporal granularity. A common paradigm in existing Video LLMs is to encode video frames into a flat sequence of visual tokens via a visual encoder, which are then fed into the language model alongside textual input (Liu et al., 2024a; Lin et al., 2024a). While effective on short video tasks, this approach encounters limitations when scaling to long-form videos. In long-form video scenarios, densely sampled frames

---

[*]Interned at the Institute of Automation, Chinese Academy of Sciences.
[†]Corresponding authors. Emails: cjzhang@bjtu.edu.cn, jliu@nlpr.ia.ac.cn.

often result in overly long token sequences, leading to significant computational overhead and potentially exceeding the context window of LLMs. To mitigate this, recent studies (Xu et al., 2025; Huang et al., 2024b; Zhao et al., 2026) adopt the Slow-Fast (Feichtenhofer et al., 2019) architecture, which decouples video features into temporal and spatial tokens before concatenating them as input to the LLM. However, this design remains limited. Temporal and spatial information are still processed jointly within the LLM, lacking explicit separation, and often leading to temporal confusion or hallucinations (Bae et al., 2025). Moreover, spatial features are typically extracted through query-agnostic uniform sampling, which lacks alignment with the language query and may overlook critical visual information relevant to the task.

To address these limitations, we propose Divid, a novel framework that explicitly integrates spatio-temporal disentanglement inside the LLM decoder architecture. Inspired by humans first scan the global motion of a video before attending to detailed video content, Divid employs two complementary branches, as shown in Figure 1. The Temporal Branch processes densely sampled, low-resolution frames to effectively capture long-range video dynamics. Concurrently, the Spatial Branch dynamically selects a subset of high-resolution keyframes based on attention maps generated by the temporal branch, enabling the model to perform query-aware Top-K key frame selection. Furthermore, we introduce a spatio-temporal soft-router embedded within each decoder layer. This component allows each language token to adaptively integrate spatial and temporal features through a lightweight, query-conditioned gating mechanism.

To further facilitate temporally grounded video understanding, we construct TempGCap, a large-scale dataset consisting of 559K instruction-based, timestamp-grounded video-text pairs. The dataset combines human-annotated action boundaries, reassembled long-form videos, and curated stitched short videos, offering rich temporal supervision and diverse scene coverage. Compared to previous instruction-tuning datasets such as Momentor-10M (Qian et al., 2024a), TempGCap offers significantly enhanced temporal precision and improved alignment between video content and language annotations, making it particularly suitable for training video-language models on temporally grounded understanding tasks.

Extensive experiments on Charades-STA, ReXTime, CG-Bench, and NExT-GQA demonstrate that Divid consistently outperforms existing Video LLMs, achieving superior temporal grounding and video understanding. Notably, Divid achieves these enhancements while substantially reducing computational costs. In summary, our work makes the following key contributions:

- We propose Divid, a dual-branch video-language framework that disentangles spatial and temporal modeling within the LLM decoder. It incorporates temporally guided keyframe selection and a soft-router for dynamic spatio-temporal fusion, enabling fine-grained temporal grounding.
- We construct TempGCap, a large-scale instruction-tuning dataset with 559K timestamp-grounded video-text pairs, providing high temporal precision and diverse video coverage for training temporally grounded video-language models.
- Extensive experiments on multiple temporal grounding and grounded VideoQA benchmarks demonstrate the effectiveness and generalization ability of our approach across diverse temporal understanding tasks.

## 2 RELATED WORK

**Video Large Language Models.** Recent progress in large language models (LLMs) (Chiang et al., 2023; Touvron et al., 2023; Bai et al., 2025) has spurred the development of video LLMs, which integrate vision encoders with LLMs via projection bridges (Li et al., 2025b; Zhang et al., 2023). However, when applied to long videos, this paradigm leads to excessively long visual token sequences that exceed LLMs' context limits and incur high computational costs. To address this limitation, several studies compress video features before passing them into the LLM (Song et al., 2024; Shen et al., 2024; Zhang et al., 2025), thereby enabling the model to process a greater number of frames within the available context window. However, such compression, particularly along the temporal dimension, negatively affects the model's temporal grounding capabilities. Inspired by the Slow-Fast paradigm (Feichtenhofer et al., 2019), several recent works (Huang et al., 2024b; Xu et al., 2025; Zhao et al., 2026; Bae et al., 2025) aim to disentangle video representations into

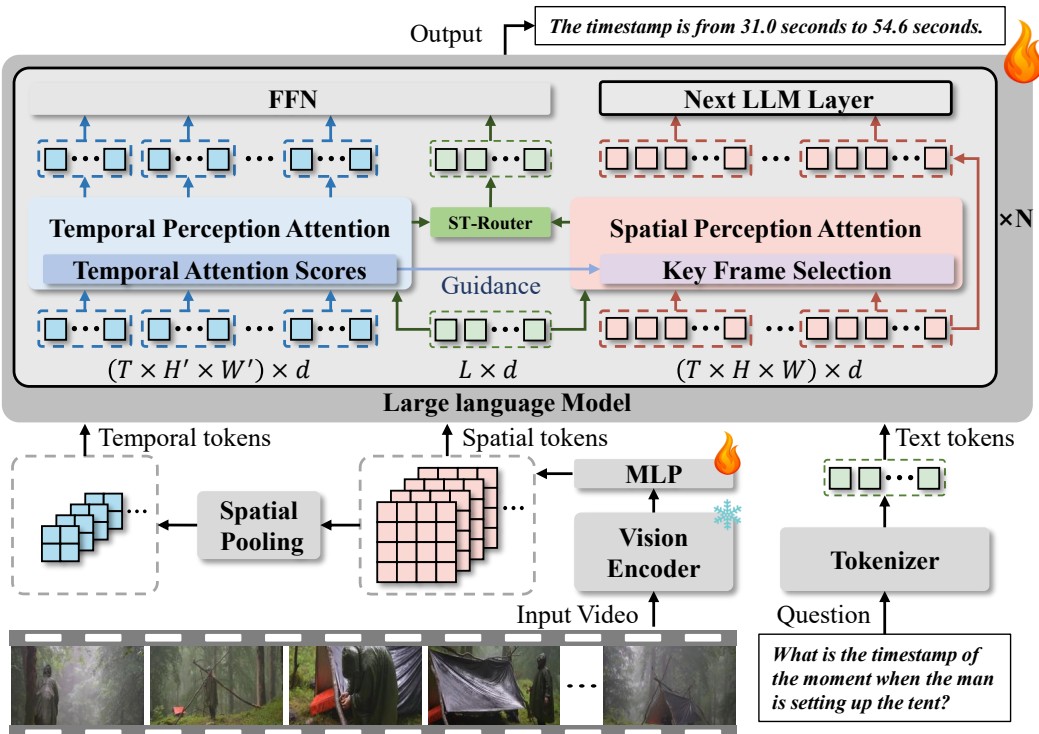

Figure 1: Overview of the proposed **Divid** framework.

fast tokens for capturing long-range temporal dependencies and slow tokens for preserving spatial details. LITA (Huang et al., 2024b) and SlowFast-VideoLLM (Xu et al., 2025) implement this idea by extracting temporal and spatial features separately, but simply concatenate them before feeding into the LLM, leading to limited spatio-temporal separation. VideoExpert (Zhao et al., 2026) introduces a Mixture-of-Experts (MoE) strategy, where temporal tokens regress timestamp boundaries via a dedicated head, while spatial tokens are used for caption generation. However, these methods still operate under a unified attention space, making them prone to spatio-temporal entanglement and hallucination (Bae et al., 2025). To mitigate this, MASH-VLM (Bae et al., 2025) proposes to decouple attention via masking, yet it relies heavily on predefined positional encodings and remains incompatible with efficient attention mechanisms. Moreover, spatial tokens in these approaches are typically derived via uniform frame sampling, which disregards query relevance and may overlook task-critical visual content.

**Temporally Grounded Video Understanding.** Temporally grounded video understanding encompasses two core tasks: *temporal grounding* and *grounded video question answering (Grounded VideoQA)*. The temporal grounding task focuses on localizing the segment of a video that semantically corresponds to a given natural language description (Gao et al., 2017). Building upon this, Grounded VideoQA requires models not only to answer questions about the video, but also to locate the relevant temporal segments that serve as evidence for their answers (Xiao et al., 2024). Existing Video LLM based approaches typically tackle the temporal grounding task via two primary strategies: (1) *timestamp-as-text generation*, where the LLM decoder directly generates the temporal boundaries as natural language text (Ren et al., 2024; Zeng et al., 2025; Chen et al., 2024c; Li et al., 2024c); and (2) *boundary regression*, where an auxiliary decoder or additional network layers are employed to regress the timestamp boundaries precisely (Liu et al., 2024d; 2025; Zhao et al., 2026). To enhance temporal grounding, recent studies introduce instruction-tuning datasets that provide timestamp-aware supervision (Huang et al., 2024a; Qian et al., 2024a; Wang et al., 2024b). These datasets aim to bridge the gap between natural language queries and their corresponding temporal segments in video. However, they often face a trade-off between annotation quality and scalability. Consequently, Video LLM based methods still exhibit limitations when applied to long-form videos and fine-grained temporal reasoning. In this work, we investigate effective strategies for constructing large-scale, high-quality instruction-tuning datasets to enhance the temporal grounding capabilities of Video LLMs.

## 3 METHOD

**Overview.** We propose *Divid*, a dual-branch framework that disentangles spatial and temporal modeling for long-form video understanding within a large language model (LLM). As shown in Figure 1, given an input video and a textual query, a vision encoder extracts frame-wise spatial features, which are projected into the language embedding space. Spatial pooling is then applied to obtain low-resolution temporal tokens, while the original high-resolution features are retained for spatial reasoning. A text encoder encodes the query into a sequence of language tokens. The core component of Divid is a disentangled spatio-temporal decoder, which comprises two parallel attention branches: the temporal perception branch and the spatial perception branch. The temporal branch processes temporally dense but spatially low-resolution features to capture global video dynamics and generates attention maps that guide the spatial branch in selecting key frames. The spatial branch then attends to high-resolution features of selected frames to extract fine-grained visual details. To integrate the complementary information from both branches, a token-level spatio-temporal soft-router is introduced. It employs a query-conditioned gating mechanism to adaptively fuse the text-side attended features from the spatial and temporal branches. The resulting fused representation serves as input to subsequent decoding layers, enabling joint spatio-temporal reasoning guided by language.

### 3.1 VIDEO AND TEXT ENCODING

Given a video $V$ composed of $T$ frames $\{I_1, I_2, \ldots, I_T\}$, each frame $I_t$ is processed by a visual encoder $f_{\text{vis}}$, followed by a multi-layer perceptron (MLP) to project the output into the shared embedding space:

$$\mathbf{X} = \{\mathbf{x}_t\}_{t=1}^T = \{\text{MLP}(f_{\text{vis}}(I_t))\}_{t=1}^T, \quad \mathbf{X} \in \mathbb{R}^{T \times H \times W \times d}, \tag{1}$$

where $H$ and $W$ denote the spatial resolution of the feature map, and $d$ is the feature dimension. Then, spatial pooling is applied to the video features to obtain the lower-resolution video features:

$$\mathbf{X}' = \{\mathbf{z}_t\}_{t=1}^T, \quad \mathbf{z}_t = f_{\text{pool}}(\mathbf{x}_t), \quad \mathbf{X}' \in \mathbb{R}^{T \times H' \times W' \times d}, \tag{2}$$

where $f_{\text{pool}}(\cdot)$ denotes a spatial average pooling operation. Without loss of generality, we assume $H' = W' = 1$, and reshape $\mathbf{X}' \in \mathbb{R}^{T \times 1 \times 1 \times d}$ into $\mathbb{R}^{T \times d}$, which we still denote as $\mathbf{X}'$. The low-resolution video features are utilized as temporal tokens $\mathbf{X}'$ for modeling temporal context, while the high-resolution video features serve as spatial tokens $\mathbf{X}$ for subsequent key frame selection and fine-grained spatial understanding.

Given a textual query $Q$ consisting of $L$ tokens, we encode it with a text encoder $f_{\text{text}}$:

$$\mathbf{X}_{\text{text}} = f_{\text{text}}(Q), \quad \mathbf{X}_{\text{text}} \in \mathbb{R}^{L \times d}, \tag{3}$$

where $f_{\text{text}}$ maps each token into the shared $d$-dimensional embedding space.

### 3.2 DISENTANGLED SPATIO-TEMPORAL LLM

The key component of our Divid framework is the disentangled spatio-temporal decoder layer, which enables the large language model (LLM) to perform spatial and temporal modeling in a decoupled yet collaborative manner. Each decoder layer consists of two parallel attention branches: the *Temporal Perception Attention* branch and the *Spatial Perception Attention* branch.

**Temporal Perception Attention.** Given the video temporal tokens $\mathbf{X}' \in \mathbb{R}^{T \times d}$ and the text tokens $\mathbf{X}_{\text{text}} \in \mathbb{R}^{L \times d}$, we concatenate them along the token dimension to form the joint input sequence $\mathbf{X}^{\text{temp}} = \text{Concat}(\mathbf{X}', \mathbf{X}_{\text{text}}) \in \mathbb{R}^{(T+L) \times d}$, where $d$ is the dimension of the token embeddings shared across both video and text modalities.

The temporal attention branch applies self-attention to model long-range temporal dependencies:

$$\mathbf{A}_{\text{temp}} = \text{Softmax}\left(\frac{(\mathbf{X}^{\text{temp}}\mathbf{W}_Q)(\mathbf{X}^{\text{temp}}\mathbf{W}_K)^{\top}}{\sqrt{d}}\right), \quad \mathbf{A}_{\text{temp}} \in \mathbb{R}^{(T+L) \times (T+L)}. \tag{4}$$

We obtain temporally-aware feature representation $\mathbf{Z}^{\text{temp}} = \mathbf{A}_{\text{temp}} \cdot (\mathbf{X}^{\text{temp}}\mathbf{W}_V)$, where $\mathbf{W}_Q, \mathbf{W}_K, \mathbf{W}_V \in \mathbb{R}^{d \times d}$ are learnable projection matrices. To guide the spatial branch, we extract the cross-modal attention map $\mathbf{A}_{\text{temp}}^{\text{T2V}} \in \mathbb{R}^{L \times T}$ from $\mathbf{A}_{\text{temp}}$, corresponding to the attention from text queries to video keys.

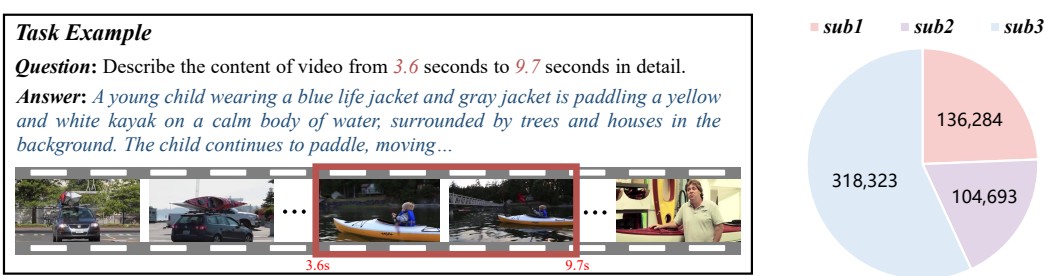

Figure 2: The Timestamp-Guided Captioning and the proposed TempGCap dataset.

**Key Frame Selection.** To reduce computational cost and focus on the most relevant visual content, we select a fixed number of key frames based on the cross-modal attention map $\mathbf{A}_{\text{temp}}^{\text{T2V}} \in \mathbb{R}^{L \times T}$, which indicates how strongly each text token attends to each video frame. Specifically, we first compute an importance score $\mathbf{s}_{t2v} \in \mathbb{R}^T$ for each video frame by averaging its attention weights across all text tokens. Then, we select the indices of the top-$k$ frames with the highest importance scores and extract the corresponding high-resolution spatial features from $\mathbf{X} \in \mathbb{R}^{T \times H \times W \times d}$:

$$\mathbf{X}_{\text{key}} = \mathbf{X}[\mathcal{I}_{\text{key}}] \in \mathbb{R}^{k \times H \times W \times d}, \quad \mathcal{I}_{\text{key}} = \text{TopK}(\mathbf{s}_{t2v}, k). \tag{5}$$

**Spatial Perception Attention.** We flatten the selected spatial features $\mathbf{X}_{\text{key}} \in \mathbb{R}^{k \times H \times W \times d}$, and apply a linear projection to map them into the feature space of the current decoder layer. After flattening and projection, we obtain the spatial tokens $\mathbf{X}^{\text{key}} \in \mathbb{R}^{L_v \times d}$, where $L_v = k \times H \times W$. Then, the spatial tokens $\mathbf{X}^{\text{key}}$ are concatenated with the text tokens $\mathbf{X}_{\text{text}}$ to form a joint input sequence $\mathbf{X}^{\text{spatial}} \in \mathbb{R}^{(L_v+L) \times d}$. This sequence is then processed through a self-attention mechanism to effectively capture fine-grained spatial dependencies and interactions between spatial and textual information. Finally, we obtain spatially-aware feature representation $\mathbf{Z}^{\text{spat}} \in \mathbb{R}^{(L_v+L) \times d}$. The self-attention parameters are shared with the temporal attention branch.

**Spatio-temporal Soft-Router.** To adaptively integrate information from both temporal and spatial branches, we introduce a token-level spatio-temporal soft-router that dynamically fuses the attended text features from the two branches through a query-dependent gating mechanism. Let $\mathbf{H}^{\text{temp}}, \mathbf{H}^{\text{spat}} \in \mathbb{R}^{L \times d}$ denote the text-token outputs extracted from the temporal and spatial branches, respectively (i.e., the text-aligned slices of $\mathbf{Z}^{\text{temp}}$ and $\mathbf{Z}^{\text{spat}}$). We use the original textual representation $\mathbf{X}_{\text{text}} \in \mathbb{R}^{L \times d}$, prior to any cross-modal attention, as the input to a lightweight gating module. This module generates a routing score for each token:

$$\mathbf{G} = \text{Softmax}(\mathbf{X}_{\text{text}}\mathbf{W}_g + \mathbf{b}_g) \in \mathbb{R}^{L \times 2}, \tag{6}$$

where $\mathbf{W}_g \in \mathbb{R}^{d \times 2}$ and $\mathbf{b}_g \in \mathbb{R}^2$ are learnable parameters. The $i$-th row of $\mathbf{G}$, denoted $\mathbf{g}_i$, represents the soft weights over the temporal and spatial branches for the $i$-th text token. We then perform a weighted combination of the branch-specific attended features:

$$\mathbf{Z}_{\mathbf{i}}^{\text{text}} = g_i^{(1)} \cdot \mathbf{H}_i^{\text{temp}} + g_i^{(2)} \cdot \mathbf{H}_i^{\text{spat}}, \quad \text{for } i = 1, \dots, L, \tag{7}$$

where $g_i^{(1)}$ and $g_i^{(2)}$ are the temporal and spatial routing weights from $\mathbf{g}_i$. The resulting fused representation $\mathbf{Z}^{\text{text}} \in \mathbb{R}^{L \times d}$ adaptively integrates spatial and temporal cues at the token level and is used to update the temporal stream in subsequent decoding layers.

### 3.3 TEMPGCAP: INSTRUCTION DATA FOR TIMESTAMP-GUIDED CAPTIONING

Recent studies (Golkar et al., 2023; Zeng et al., 2025; Jung et al., 2025) have shown that Video LLMs often struggle with temporal understanding and are prone to hallucinations. Our proposed Divid model, which directly predicts timestamps using an LLM decoder, is particularly prone to such errors. Moreover, as Divid framework leverages temporal attention to guide spatial selection, strong temporal perception is essential for accurate grounding. To address this issue, we introduce a **timestamp-guided captioning** task. As illustrated in Figure 2, this task requires the model to generate a natural language description for a specific temporal segment of a video (e.g., from the 3rd to the 5th second). The model must accurately localize the described segment while avoiding interference from irrelevant content outside the target window. This task is designed to enhance the model's

temporal grounding capability and to promote a more precise understanding of temporal structure in videos. However, existing timestamp-guided captioning datasets often struggle to balance scale and annotation quality. For example, Momentor (Qian et al., 2024a) provides 1.7M timestamp-guided captioning instances, but the alignment between video segments and textual descriptions is often imprecise. Additionally, the dataset is sourced from only around 65K YouTube videos, which limits content diversity. In contrast, the high-quality timestamp-guided captioning subset used in VTimeLLM-Stage2 (Huang et al., 2024a) and InternVid-G (Wang et al., 2024e) contains only about 95K samples, which is insufficient for learning generalizable temporal grounding capabilities.

To this end, we construct a new instruction-tuning dataset, **TempGCap**, to support the proposed timestamp-guided captioning task. The dataset is built from three complementary sources, as shown in Figure 2. **Sub1: Reannotating untrimmed videos with manual temporal annotations.** We first adapt existing temporal grounding datasets to align with our task setting. Additionally, we re-annotate action detection datasets (e.g., HACS (Zhao et al., 2019)), which typically include coarse action categories and their temporal boundaries but lack natural language descriptions. To obtain fine-grained captions, we utilize the advanced video captioning model Tarsier (Wang et al., 2024c) to generate detailed descriptions for each annotated segment. **Sub2: Recovering untrimmed contexts from captioned clips.** We collect the original untrimmed videos corresponding to captioned clips with high-quality textual annotations. Using the original timestamps of these trimmed segments, we construct timestamp-guided captioning samples. To improve annotation precision and diversity, we further refine the temporal boundaries by measuring visual similarity between boundary frames and their adjacent frames. **Sub3: Synthesizing pseudo-untrimmed videos.** When original untrimmed videos are unavailable, we synthesize pseudo-untrimmed videos by concatenating several short clips into a longer video sequence. We then select one of the original clips and use its associated caption along with its temporal location within the sequence to form a timestamp-guided captioning instance. After filtering and random quality inspection, we construct a high-quality instruction-tuning dataset comprising **559K** timestamp-guided captioning samples, sourced from **428K** videos, which ensures wide coverage across diverse video domains. Additional details are included in the appendix.

## 4 EXPERIMENT

### 4.1 IMPLEMENTATION DETAILS

We use ViT-g/14 from EVA-CLIP (Sun et al., 2023) as the vision encoder and Qwen2-7B (Yang et al., 2024) as the large language model. In the first pre-training stage, we use LLaVA-558K dataset (Liu et al., 2023) with WebVid videos (Bain et al., 2021). Following (Chen et al., 2024c), we prepend each frame with a textual token indicating its timestamp (e.g., "3.0 second"), where the textual tokens are directly encoded using the tokenizer of the LLM. During this stage, only the multi-modal projector and the LLM layer projectors of spatial tokens are updated. The learning rate is set to $1 \times 10^{-3}$, and the batch size is 256. For instruction tuning, we combine our TempG-Cap dataset (559k samples) with 77K timestamp-grounded samples from DideMo (Anne Hendricks et al., 2017) and ActivityNet Captions (Krishna et al., 2017). In addition, we incorporate LLaVA-Video-178K (Zhang et al., 2024c) and DeVE-QA (Qin et al., 2025) for video QA instruction learning. The learning rate during this stage is set to $2 \times 10^{-5}$, and the batch size is 128. Throughout all training stages, the visual encoder is kept frozen. In our implementation, we adopt practical configurations to balance temporal coverage and spatial resolution. Specifically, the temporal attention branch operates on $T = 128$ frames with a reduced spatial resolution of $H' \times W' = 16$, enabling efficient modeling of long-range dynamics. For the spatial attention branch, we select $k = 32$ key frames based on temporal guidance, and retain high-resolution features with $H \times W = 64$ to preserve fine-grained spatial details. Please refer to Appendix for additional hyper-parameters.

### 4.2 EVALUATION SETUPS

**Temporal Grounding.** This task aims to localize a temporal segment within a video that corresponds to a given language query, by predicting the start and end timestamps. We adopt the Charades-STA (Gao et al., 2017) dataset as the evaluation benchmark, which consists of 6.6K videos and 16.1K query-moment pairs. The average durations of the videos and target moments are 30.6 seconds and 8.1 seconds, respectively. Following the zero-shot setting, we evaluate the model on the

Table 1: Temporal Grounding on Charades-STA (Gao et al., 2017) Dataset.

| Method | Size | R@0.5 | R@0.7 | mIoU |
|---|---|---|---|---|
| (FT) Moment-DETR (Lei et al., 2021) | – | 52.1 | 30.6 | 45.5 |
| (FT) UniVTG (Lin et al., 2023) | – | 58.1 | 35.6 | 50.1 |
| (FT) R$^2$-Tuning (Liu et al., 2024c) | – | 59.8 | 37.0 | 50.9 |
| GPT-4o | – | – | – | 35.7 |
| Grounded-VideoLLM (Wang et al., 2024b) | 4B | 36.4 | 19.7 | 36.8 |
| E.T. Chat (Liu et al., 2024d) | 4B | 45.9 | 20.0 | 42.3 |
| Qwen2.5-VL (Bai et al., 2025) | 3B | – | – | 38.8 |
| VTimeLLM (Huang et al., 2024a) | 13B | 34.3 | 14.7 | 34.6 |
| Qwen2.5-VL (Bai et al., 2025) | 72B | – | – | 50.9 |
| Qwen2.5-VL (Bai et al., 2025) | 7B | – | – | 43.6 |
| TimeChat (Ren et al., 2024) | 7B | 32.2 | 13.4 | – |
| Momentor (Qian et al., 2024a) | 7B | 26.6 | 11.6 | 28.5 |
| HawkEye (Wang et al., 2024e) | 7B | 31.4 | 14.5 | 33.7 |
| ChatVTG (Qu et al., 2024) | 7B | 33.0 | 15.9 | 34.9 |
| VideoChat-TPO (Yan et al., 2025) | 7B | 40.2 | 18.4 | 38.1 |
| VideoChat-T (Zeng et al., 2025) | 7B | 48.7 | 24.0 | – |
| VideoChat-Flash (Li et al., 2024c) | 7B | 53.1 | 27.6 | – |
| TimeMarker (Chen et al., 2024c) | 8B | 51.9 | 26.9 | 48.4 |
| TimeSearch (Pan et al., 2025) | 7B | 52.4 | 24.5 | 48.6 |
| VideoMind (Liu et al., 2025) | 1.5B | 51.1 | 26.0 | 45.2 |
| VideoMind (Liu et al., 2025) | 7B | 59.1 | 31.2 | 50.2 |
| **Divid (Ours)** | 1.5B | 51.4 | 26.9 | 47.3 |
| **Divid (Ours)** | 7B | 59.5 | 31.3 | 51.3 |

Table 2: Grounded VideoQA on CG-Bench (Chen et al., 2025).

| Method | Size | mIoU | R@IoU | A@IoU |
|---|---|---|---|---|
| GPT-4o | – | 5.62 | 8.30 | **4.38** |
| GPT-4o-mini | – | 3.75 | 5.18 | 2.21 |
| Gemini-1.5-Pro | – | 3.95 | 5.81 | 2.53 |
| Gemini-1.5-Flash | – | 3.67 | 5.44 | 2.45 |
| Claude-3.5-Sonnet | – | 3.99 | 5.67 | 2.79 |
| Qwen2-VL (Wang et al., 2024d) | 72B | 3.58 | 5.32 | 3.31 |
| VITA (Fu et al., 2024) | 8×7B | 3.06 | 3.53 | 2.06 |
| ShareGPT4Video (Chen et al., 2024b) | 16B | 1.85 | 2.65 | 1.01 |
| Video-CCAM (Fei et al., 2024) | 14B | 2.63 | 3.48 | 1.83 |
| Chat-UniVi-1.5 (Jin et al., 2024) | 13B | 2.07 | 2.53 | 1.21 |
| LLaVA-OV (Li et al., 2024a) | 13B | 1.63 | 1.78 | 1.08 |
| Video-LLaVA (Lin et al., 2024a) | 7B | 1.13 | 1.96 | 0.59 |
| VideoLLaMA (Zhang et al., 2023) | 7B | 1.21 | 1.87 | 0.84 |
| Videochat2 (Li et al., 2024b) | 7B | 1.28 | 1.98 | 0.94 |
| Qwen-VL-Chat (Bai et al., 2023) | 7B | 0.89 | 1.19 | 0.42 |
| ST-LLM (Liu et al., 2024b) | 7B | 2.23 | 2.86 | 1.13 |
| ViLA (Lin et al., 2024b) | 8B | 1.56 | 2.89 | 1.35 |
| MiniCPM-v2.6 (Yao et al., 2024) | 8B | 2.35 | 2.61 | 1.04 |
| LongVA (Zhang et al., 2024b) | 7B | 2.94 | 3.86 | 1.78 |
| Kangaroo (Liu et al., 2026) | 8B | 2.56 | 2.81 | 1.94 |
| InternVL2 (Chen et al., 2024d) | 7B | 3.91 | 5.05 | 2.64 |
| **Divid (Ours)** | 1.5B | 4.77 | 6.40 | 2.37 |
| **Divid (Ours)** | 7B | **5.74** | **8.36** | 4.11 |

Table 3: Grounded VideoQA on NExT-GQA (Xiao et al., 2024).

| Method | LLM Size | IoU | | | IoP | | | Acc@GQA |
|---|---|---|---|---|---|---|---|---|
| | | R@0.3 | R@0.5 | mIoU | R@0.3 | R@0.5 | mIoP | |
| FrozenBiLM NG+ (Yang et al., 2022) | 890M | 13.5 | 6.1 | 9.6 | 28.5 | 23.7 | 24.2 | 17.5 |
| VIOLETv2 (Fu et al., 2023) | – | 4.3 | 1.3 | 3.1 | 25.1 | 23.3 | 23.6 | 12.8 |
| SeViLA (Yu et al., 2023) | 4B | 29.2 | 13.8 | 21.7 | 34.7 | 22.9 | 29.5 | 16.6 |
| LangRepo (Kahatapitiya et al., 2025) | 8×7B | – | 12.2 | 18.5 | – | 28.7 | 31.3 | 17.1 |
| VideoStreaming (Qian et al., 2024b) | 8.3B | – | 13.3 | 19.3 | – | 31.0 | 32.2 | 17.8 |
| LLoVi (Zhang et al., 2024a) | 1.8T | – | 15.3 | 20.0 | – | 36.9 | 37.3 | 24.3 |
| HawkEye (Wang et al., 2024e) | 7B | 37.0 | 19.5 | 25.7 | – | – | – | – |
| Grounded-VideoLLM (Wang et al., 2024b) | 4B | 30.2 | 18.0 | 21.1 | 42.6 | 34.4 | 34.5 | 26.7 |
| VideoChat-TPO (Yan et al., 2025) | 7B | 41.2 | 23.4 | 27.7 | 47.5 | 32.8 | 35.6 | 25.5 |
| VideoMind (Liu et al., 2025) | 1.5B | 45.2 | 23.2 | 28.6 | 51.3 | 32.6 | 36.4 | 25.2 |
| VideoMind (Liu et al., 2025) | 7B | 50.2 | 25.8 | 31.4 | **56.0** | 35.3 | 39.0 | 28.2 |
| **Divid (Ours)** | 1.5B | 47.5 | 26.8 | 32.9 | 52.0 | 33.7 | 38.9 | 26.4 |
| **Divid (Ours)** | 7B | **51.3** | **27.5** | **34.5** | 54.6 | **38.2** | **40.8** | **29.2** |

standard 3.7K test queries. For evaluation, we report Recall@1 under different Intersection-over-Union (IoU) thresholds, including R1@0.5 and R1@0.7, along with the mean IoU (mIoU).

**Grounded Video Question Answering.** This task requires the model to not only answer questions about video content but also to provide grounded evidence by localizing the relevant temporal segments. We evaluate our approach on three representative datasets: *CG-Bench* (Chen et al., 2025), *NExT-GQA* (Xiao et al., 2024) and *ReXTime* (Chen et al., 2024a). For CG-Bench, we report three metrics. Mean IoU (mIoU) measures the average overlap between predicted and ground-truth time intervals. Rec.@IoU (R@IoU) computes the average recall at IoU thresholds from 0.1 to 0.5, reflecting how well the model retrieves relevant segments. Acc.@IoU (A@IoU) combines answer correctness with grounding quality: a prediction is correct only if the answer is right and the predicted interval has tIoU $> 0$ with the ground truth. For NExT-GQA, evaluation includes two parts. IoU and IoP are used to assess whether the predicted temporal segments align with the annotated evidence. Acc@GQA further requires the answer to be correct and the predicted interval to have IoP $\geq 0.5$ with the ground truth, reflecting grounding consistency. ReXTime evaluates models using mean IoU (mIoU) and answer accuracy (Acc@IoU) under the constraint of IoU $\geq 0.5$.

## 4.3 MAIN RESULTS

**Performance on Temporal Grounding.** As shown in Table 1, our proposed method Divid achieves strong performance on the Charades-STA benchmark under the zero-shot setting. The rows labeled with (FT) indicate models that are fine-tuned on the downstream training set. We first examine the performance of the 1.5B version. Despite being a lightweight model, Divid (1.5B) achieves 51.4 in R@0.5 and 47.3 in mIoU, outperforming most zero-shot baselines of similar or even larger size. Notably, Divid (1.5B) exceeds the performance of many larger 7B models. For example, it surpasses Momentor (7B), HawkEye (7B), TimeChat (7B), and ChatVTG (7B) by margins ranging from 12.4 to 18.8 in mIoU. In particular, Divid (1.5B) outperforms Momentor (7B) by 18.8 in mIoU (47.3 vs. 28.5), while also achieving higher R@0.5 (51.4 vs. 26.6). These comparisons emphasize the

Table 4: Grounded VideoQA on ReXTime (Chen et al., 2024a).

| Method | LLM Size | FT | R@0.3 | R@0.5 | mIoU | Acc@IoU |
|---|---|---|---|---|---|---|
| VTimeLLM (Huang et al., 2024a) | 7B | ✓ | 43.69 | 26.13 | 29.92 | 17.13 |
| TimeChat (Ren et al., 2024) | 7B | ✓ | 40.13 | 21.42 | 26.29 | 10.92 |
| VTimeLLM (Huang et al., 2024a) | 7B | – | 28.84 | 17.41 | 20.14 | – |
| TimeChat (Ren et al., 2024) | 7B | – | 14.42 | 7.61 | 11.65 | – |
| LITA (Huang et al., 2024b) | 13B | – | 29.49 | 16.29 | 21.49 | – |
| VideoMind (Liu et al., 2025) | 1.5B | – | 34.31 | 22.69 | 24.83 | 17.26 |
| VideoMind (Liu et al., 2025) | 7B | – | 38.22 | 25.52 | 27.61 | 20.20 |
| **Divid (Ours)** | 1.5B | – | 40.50 | 26.82 | 29.71 | 18.57 |
| **Divid (Ours)** | 7B | – | **42.56** | **31.05** | **35.78** | **22.26** |

Table 5: Comparison with existing frameworks.

| Method | LLM TFLOPs | Charades-STA | | ReXTime | |
|---|---|---|---|---|---|
| | | R1@0.5 | mIoU | mIoU | Acc@IoU |
| Full | 28.2 | 51.55 | 47.76 | 30.16 | 19.11 |
| Slow-Fast | 16.2 | 50.65 | 46.71 | 29.49 | 18.45 |
| Divid | 10.5 | 51.37 | 47.33 | 29.71 | 18.57 |

Table 6: The impact of temporal guidance.

| Method | Charades-STA | | ReXTime | |
|---|---|---|---|---|
| | R1@0.5 | mIoU | mIoU | Acc@IoU |
| Uniform | 49.70 | 46.27 | 27.61 | 17.32 |
| Weighted | 48.62 | 45.43 | 26.44 | 16.78 |
| Segment | 50.96 | 47.20 | 28.82 | 17.59 |
| Importance | 50.86 | 47.10 | 28.83 | 17.92 |
| Top-K | 51.37 | 47.33 | 29.71 | 18.57 |

efficiency and effectiveness of our model design, showing that careful architectural choices, together with high-quality timestamp-guided supervision provided by the TempGCap dataset, can compensate for and even surpass the benefit of larger model size. The 7B version of Divid further improves performance, achieving 59.5 in R@0.5, 31.3 in R@0.7, and 51.3 in mIoU. Compared to TimeMarker (8B), which is trained on 45 million video-text samples, Divid (7B) achieves higher performance by 7.6 in R@0.5, 4.4 in R@0.7, and 2.9 in mIoU. This demonstrates not only the strength of our model architecture, but also the effectiveness of the TempGCap dataset in providing fine-grained, instruction-aligned temporal supervision that supports precise grounding. Furthermore, Divid (7B) outperforms Qwen2.5-VL (72B) in mIoU by 0.4, despite having only one-tenth of the parameters.

**Performance on Grounded VideoQA.** As shown in Table 2, our proposed method Divid achieves state-of-the-art performance among open-source models on the CG-Bench benchmark under the zero-shot setting. Our 1.5B model already demonstrates competitive results, achieving 4.77 in mIoU, 6.40 in R@IoU, and 2.37 in A@IoU. These results surpass several larger models, including Video-LLaVA (7B), ST-LLM (7B), and VideoLLaMA (7B), demonstrating the effectiveness of our architecture in handling grounded video question answering with limited model capacity. With parameter scaling, the performance of Divid further improves. Our 7B model achieves 5.74 in mIoU, 8.36 in R@IoU, and 4.11 in A@IoU, setting a new state-of-the-art among all open-source models. Compared to Qwen2-VL (72B), which is a much larger model, our 7B version still achieves higher mIoU, higher R@IoU, and higher A@IoU. When compared with commercial models, Divid (7B) achieves comparable or even superior performance across most metrics. It surpasses Gemini-1.5-Pro, Claude-3.5-Sonnet, and GPT-4o-mini on all three metrics. While GPT-4o achieves slightly higher A@IoU, Divid (7B) matches or exceeds GPT-4o in mIoU and R@IoU. These results highlight the strong temporal grounding capability of our framework, demonstrating that with effective spatio-temporal modeling and the proposed instruction-tuning dataset TempGCap, Divid is able to compete with and even outperform larger-scale or commercial-grade models in grounded video understanding. As shown in Table 3 and Table 4, Divid achieves state-of-the-art zero-shot performance on both NExT-GQA and ReXTime. The 1.5B model already outperforms most baselines, while the 7B version further improves to 34.5 mIoU and 29.2 Acc@GQA on NExT-GQA, and 35.78 mIoU and 22.26 Acc@IoU on ReXTime, surpassing VideoMind (7B) by a large margin. These results confirm Divid's effectiveness in fine-grained evidence localization and temporal question answering.

## 4.4 ABLATION STUDIES

To better understand the contribution of each component in our *Divid* framework, we evaluate on Charades-STA (Gao et al., 2017) for temporal grounding and ReXTime (Chen et al., 2024a) for grounded VideoQA. Unless otherwise specified, we use ViT-g/14 from EVA-CLIP (Sun et al., 2023) as the vision encoder and Qwen2-1.5B from (Wang et al., 2024d) as the large language model.

Table 7: Ablation studies of ST-Router.

| Method | Charades-STA | | ReXTime | |
|---|---|---|---|---|
| | R1@0.5 | mIoU | mIoU | Acc@IoU |
| Add | 50.66 | 46.81 | 29.11 | 18.23 |
| Weight | 50.98 | 47.13 | 29.31 | 18.42 |
| Soft-Router | 51.37 | 47.33 | 29.71 | 18.57 |

Table 8: Ablation studies of datasets.

| Method | Charades-STA | | ReXTime | |
|---|---|---|---|---|
| | R1@0.5 | mIoU | mIoU | Acc@IoU |
| Momentor-1M (Qian et al., 2024a) | 48.96 | 44.90 | 26.77 | 15.43 |
| Momentor (Qian et al., 2024a) | 50.47 | 46.42 | 28.75 | 16.78 |
| TempGCap | 51.37 | 47.33 | 29.71 | 18.57 |

**Ablation studies of the framework.** We compare three architectural variants: *Full*, *Slow-Fast*, and our proposed *Divid*, as shown in Table 5. The Full variant adopts the standard LLaVA-style pipeline, where all video frames are retained at high spatial resolution and directly fed into the LLM, resulting in the highest computational cost. The Slow-Fast variant encodes video into separate temporal and spatial token streams, which are concatenated before being input into the LLM. This design reduces computational cost by limiting the number of high-resolution frames while preserving temporal diversity. In contrast, our proposed *Divid* disentangles temporal and spatial modeling inside the LLM. Instead of direct concatenation, Divid leverages temporal attention to guide keyframe selection and fuses the two branches through a lightweight soft-router. This structured design improves mIoU from 46.71 to 47.33 on Charades-STA and from 29.49 to 29.71 on ReXTime. Moreover, while achieving comparable performance to the Full model, Divid operates at significantly lower computational cost (10.5 TFLOPs compared to 28.2 TFLOPs), demonstrating an effective balance between accuracy and efficiency.

**Ablation studies of the temporal guidance.** We compare five strategies for temporally guided keyframe selection, as reported in Table 6. The Uniform method samples frames at equal intervals without exploiting temporal attention, which leads to limited temporal awareness and may miss task-relevant moments. The Weighted method divides the video into $K$ segments and performs attention-based pooling within each segment, but the averaging operation tends to blur fine-grained cues, resulting in suboptimal performance. The Segment method improves upon this by explicitly selecting representative frames from different video chunks, thereby ensuring broader temporal coverage and achieving better results than Uniform and Weighted. The Importance method further incorporates attention-based probabilistic sampling to emphasize highly attended frames, which provides slight gains but still lacks robustness due to stochasticity in selection. Finally, the Top-K strategy directly chooses the $K$ frames with the highest attention scores, explicitly preserving the most salient frames for spatial reasoning. This approach achieves the best overall performance on both Charades-STA and ReXTime, improving mIoU from 46.27 to 47.33 and from 27.61 to 29.71 compared to Uniform. These results demonstrate that selective and query-aware keyframe selection is crucial for enhancing temporal grounding.

**Ablation studies of the Soft-Router.** We evaluate three strategies for fusing temporal and spatial features on the text side. The Add variant directly sums the outputs from the temporal and spatial attention branches. The Weight variant introduces a learnable scalar to balance the contribution of each branch. Soft-Router uses the original pre-attention textual features as conditioning input to dynamically compute token-wise routing weights, allowing each token to adaptively determine the proportion of temporal and spatial information. As shown in Table 7, the Soft-Router achieves the best results on both Charades-STA and ReXTime. Compared to Add, it improves mIoU by 0.52 and 0.60 respectively. The learnable Weight variant provides a slight improvement over Add, but it applies a global weighting across all tokens, which limits its flexibility. In contrast, the Soft-Router enables fine-grained control over spatio-temporal fusion at the token level, leading to more accurate reasoning and better performance. These results highlight the benefit of conditional, token-wise fusion for complex video-language alignment.

**Ablation studies of the datasets.** We study the impact of different instruction-tuning datasets on model performance by replacing TempGCap with subsets or alternatives from the Momentor dataset. As shown in Table 8, using 1 million segment-level captioning samples from Momentor (denoted as Momentor-1M) leads to a notable performance drop, with mIoU decreasing to 44.90 on Charades-STA and 26.77 on ReXTime. Expanding to the full Momentor-10M dataset brings some improvement, but still lags behind our TempGCap by 0.91 in mIoU on Charades-STA and 0.96 on ReXTime. We attribute the superiority of TempGCap to its higher annotation quality and broader video coverage. While Momentor primarily consists of auto-generated captions from a limited set of YouTube videos, TempGCap combines manually aligned boundaries, fine-grained action descriptions, and curated long-form content from diverse sources. This results in more precise temporal supervision

and better generalization across temporal grounding tasks. These results highlight the importance of both data scale and quality in training temporally grounded video-language models.

Additional results and analyses are provided in the **Appendix**.

## 5 CONCLUSION

In this paper, we introduce *Divid*, a dual-branch video-language framework that disentangles spatial and temporal modeling within a large language model. By integrating temporal perception, query-aware keyframe selection, and token-level spatio-temporal fusion, Divid enables more accurate and efficient temporally grounded video understanding. We further construct *TempGCap*, a large-scale instruction-tuning dataset with timestamp-grounded captions that provides fine-grained temporal supervision across diverse video domains. Extensive experiments on multiple benchmarks, including Charades-STA, NExT-GQA, ReXTime, and CG-Bench, demonstrate that Divid achieves state-of-the-art performance.

## 6 LIMITATIONS AND FUTURE WORK

Our framework relies on temporal attention to guide spatial selection, which may be influenced by biases inherited from the pretrained language model. To address this, future work will explore debiasing strategies for temporal attention and expand the instruction-tuning dataset with more diverse and temporally aligned annotations to further improve model generalization.

## 7 ACKNOWLEDGEMENTS

This research is supported by Artificial Intelligence-National Science and Technology Major Project (2023ZD0121200), the National Natural Science Foundation of China (62476021, 62531026, 62437001), the Strategic Priority Research Program of Chinese Academy of Sciences under Grant XDB1350103, and the Fundamental Research Funds for the Central Universities (2025JBZX062).

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

APPENDIX

## A MORE RESULTS AND ANALYSIS

Table 9: Ablation on Attention Parameter Sharing.

| Method | Charades-STA | | ReXTime | |
|---|---|---|---|---|
| | R1@0.5 | mIoU | mIoU | Acc@IoU |
| share | 51.37 | 47.33 | 29.71 | 18.57 |
| separate | 51.18 | 47.38 | 29.24 | 18.89 |

**Ablation on Attention Parameter Sharing.** We share the Q/K/V/O projection matrices between the temporal and spatial attention branches to ensure representation consistency and reduce parameter overhead. As both branches operate on joint video-text sequences, sharing attention weights promotes consistent cross-modal alignment patterns and simplifies model design. This also benefits the token-level fusion performed by the soft-router. To validate this design choice, we conducted an ablation study where we introduced LoRA-adapted, branch-specific Q/K/V/O parameters for each attention branch. This setup allows us to isolate the effect of parameter sharing while keeping the architecture and training cost nearly unchanged. As shown in Table 9, the variant with separate parameters leads to only marginal differences in performance across benchmarks. These results suggest that parameter sharing strikes a favorable balance between efficiency, representational coherence, and empirical performance.

Table 10: Ablation study on temporal and spatial branches.

(a) Temporal resolution ablation.

| Temp. Res. | Charades-STA | | ReXTime | |
|---|---|---|---|---|
| | R1@0.5 | mIoU | mIoU | Acc@IoU |
| 64 | 49.53 | 46.01 | 28.02 | 17.11 |
| 128 | 51.37 | 47.33 | 29.71 | 18.57 |
| 256 | 51.57 | 47.52 | 30.35 | 19.12 |

(b) Spatial resolution in temporal branch.

| Spatial Res. | Charades-STA | | ReXTime | |
|---|---|---|---|---|
| | R1@0.5 | mIoU | mIoU | Acc@IoU |
| 3×3 | 50.26 | 46.71 | 28.88 | 17.48 |
| 4×4 | 51.37 | 47.33 | 29.71 | 18.57 |
| 5×5 | 51.68 | 47.76 | 30.51 | 19.47 |

(c) Number of selected keyframes.

| Keyframes | Charades-STA | | ReXTime | |
|---|---|---|---|---|
| | R1@0.5 | mIoU | mIoU | Acc@IoU |
| 16 | 49.83 | 46.46 | 28.47 | 17.22 |
| 32 | 51.37 | 47.33 | 29.71 | 18.57 |
| 64 | 51.42 | 47.49 | 29.97 | 18.81 |

(d) Spatial resolution of keyframes.

| Spatial Res. | Charades-STA | | ReXTime | |
|---|---|---|---|---|
| | R1@0.5 | mIoU | mIoU | Acc@IoU |
| 8×8 | 51.37 | 47.33 | 29.71 | 18.57 |
| 12×12 | 51.39 | 47.55 | 29.90 | 18.79 |

**Temporal and Spatial branch setting.** We conduct comprehensive ablation studies to investigate the influence of temporal and spatial resolutions as well as the number of selected keyframes. As shown in Table 10 (a), increasing the temporal input length from 64 to 128 frames leads to significant gains on both Charades-STA and ReXTime, while further scaling to 256 frames brings only marginal improvements with considerable computational overhead. Table 10 (b) evaluates the effect of spatial resolution in the temporal branch, where a $4 \times 4$ resolution provides a favorable balance between accuracy and efficiency, and higher resolutions offer limited additional benefit. Regarding the number of selected keyframes, Table 10 (c) shows that using 32 frames is sufficient to capture representative visual context, with more frames yielding diminishing returns. Finally, Table 10 (d) reports results on the spatial resolution of keyframes, where increasing from $8 \times 8$ to $12 \times 12$ slightly improves performance but at the cost of higher computation. Based on these studies, we adopt 128 frames at $4 \times 4$ resolution for the temporal branch, and 32 keyframes at $8 \times 8$ resolution for the spatial branch as our default configuration, achieving strong performance while maintaining efficiency.

**Ablation on Visual Encoders.** We evaluate the impact of different visual encoders and their fine-tuning strategies on temporal grounding performance. In this experiment, *frozen* indicates that the visual encoder remains fixed during training, whereas *FT* denotes that the encoder is fine-tuned during Stage II. Comparing the first two rows of Table 11, we observe that ViT-g/14 and SigLIP achieve highly similar performance when both are frozen. Although ViT-g/14 contains a substantially larger number of parameters (approximately 1B), it is pretrained at a lower input resolution ($224 \times 224$),

Table 11: Ablation of visual encoder. FT denotes fine-tuning.

| Method | Charades-STA | | ReXTime | |
|---|---|---|---|---|
| | R1@0.5 | mIoU | mIoU | Acc@IoU |
| ViT-g/14 (frozen) | 51.37 | 47.33 | 29.71 | 18.57 |
| SigLIP (frozen) | 51.08 | 47.09 | 29.45 | 18.32 |
| SigLIP (FT) | **52.16** | **47.77** | **29.98** | **18.79** |

whereas SigLIP is pretrained at $384 \times 384$. As a result, the two encoders perform comparably on Charades-STA and ReXTime. Next, comparing the frozen and fine-tuned SigLIP settings, we find that enabling fine-tuning during Stage II leads to consistent improvements across all metrics: +1.08 R1@0.5 and +0.68 mIoU on Charades-STA, as well as +0.53 mIoU and +0.47 Acc@IoU on ReXTime. Overall, these results show that while a frozen encoder already offers strong performance, fine-tuning the visual encoder can further enhance fine-grained temporal grounding when computational resources permit.

Table 12: Ablation of fusion methods and placements.

| Method | Charades-STA | | ReXTime | |
|---|---|---|---|---|
| | R1@0.5 | mIoU | mIoU | Acc@IoU |
| Output-level | 45.80 | 42.51 | 26.44 | 16.63 |
| Block-level (4 layers) | 49.66 | 45.98 | 28.23 | 17.24 |
| Block-level (2 layers) | 50.37 | 47.04 | 29.63 | 18.20 |
| Head-fusion | 50.41 | 46.95 | 29.34 | 18.21 |
| FFN-fusion | 50.77 | 47.12 | 29.58 | 18.30 |
| Ours | 51.37 | 47.33 | 29.71 | 18.57 |

**Ablation of Fusion Methods and Placements.** Table 12 reports the ablation results on different fusion strategies and their placements within the decoder. Output-level performs fusion only at the final decoder output, which leads to the weakest results since spatial and temporal information interact only once at the end. Block-level (2 layers) and Block-level (4 layers) perform fusion every 2 or 4 decoder layers, respectively; reducing the fusion frequency clearly harms performance, demonstrating that frequent interaction between spatial and temporal cues is important. We further compare finer-grained fusion placements. Head-fusion fuses spatial–temporal features after the QKV projections but before the output projection of attention, which results in a noticeable performance drop—likely because it disrupts the attention head structure and misaligns Q/K/V distributions. FFN-fusion applies fusion after the feed-forward network, yielding better results than Head-fusion but still slightly inferior to our method. Overall, our approach, which performs fusion after attention and before the FFN at every decoder layer, achieves the best performance across all metrics. This design allows spatial and temporal information to interact fully within each layer while preserving the stability and structure of the attention mechanism, leading to consistently superior temporal grounding accuracy.

Table 13: Performance comparison on general VideoQA benchmarks.

| Method | Venue | MSVD-QA | | MSRVTT-QA | | ActivityNet-QA | |
|---|---|---|---|---|---|---|---|
| | | Acc | Score | Acc | Score | Acc | Score |
| LLaVA-Mini (Zhang et al., 2025) | ICLR25 | 70.9 | 4.0 | 59.5 | 3.6 | 53.5 | 3.5 |
| MASH-VLM (Bae et al., 2025) | CVPR25 | 74.4 | 4.0 | 61.9 | 3.6 | 49.3 | 3.4 |
| TOGA (Gupta et al., 2025) | ICCV25 | 73.8 | 3.9 | – | – | 52.0 | 3.4 |
| Ours | – | 75.5 | 4.1 | 61.2 | 3.6 | 51.8 | 3.5 |

**Generalization to VideoQA Benchmarks.** Our work is primarily motivated by advancing temporally grounded video understanding, which poses unique challenges that are not fully addressed by existing video-language models. To this end, we design both our architecture (Divid) and our instruction-tuning dataset (TempGCap) with a focus on spatio-temporal disentanglement and fine-grained timestamp supervision. Consequently, our main evaluation is conducted on temporally grounded benchmarks such as Charades-STA, ReXTime, CG-Bench, and NExT-GQA, which directly reflect our target problem setting. To further assess generalization beyond grounding-oriented tasks, we also evaluate Divid on widely used VideoQA benchmarks including MSVD-QA, MSRVTT-QA, and ActivityNet-QA. As shown in Table 13, Divid achieves competitive performance compared with recent state-of-the-art VideoLLMs such as LLaVA-Mini, MASH-VLM, and TOGA. This behavior is consistent with our design choice. Divid's dual-branch decoder, query-guided

keyframe selection, and token-level spatio-temporal routing are optimized for long-range temporal reasoning rather than short-video appearance recognition. In contrast, MSVD-QA, MSRVTT-QA, and ActivityNet-QA primarily evaluate object presence, scene attributes, and short-range actions, with limited emphasis on multi-second temporal dependencies. Therefore, the strengths of our disentangled temporal modeling do not directly translate into large gains on these appearance-centric datasets, a trend also observed in grounding-oriented VideoLLMs such as LLaVA-ST (Li et al., 2025a) and TOGA (Gupta et al., 2025). Furthermore, TempGCap is constructed to provide timestamp-grounded supervision that benefits temporal grounding tasks but is not specifically aligned with the objectives of generic VideoQA benchmarks. Despite this specialization, Divid still achieves competitive performance on these VideoQA datasets, demonstrating that its targeted design for temporally grounded long-video reasoning maintains solid generalization to broader video understanding scenarios.

Table 14: Performance comparison on Charades-STA, finetuning on the training set.

| Method | LLM Size | R@0.5 | R@0.7 | mIoU |
|---|---|---|---|---|
| VideoChat-R1 (Li et al., 2025c) | 7B | 71.7 | 50.2 | 60.8 |
| TimeZero (Wang et al., 2025) | 7B | 72.5 | 47.9 | – |
| Ours | 7B | 72.6 | 50.8 | 61.3 |

**Comparison with more advanced methods.** To further validate the effectiveness of our model architecture and the TempGCap dataset, we compare our 7B model with two advanced temporal grounding systems, VideoChat-R1 (Li et al., 2025c) and TimeZero (Wang et al., 2025), both of which are fine-tuned on the Charades-STA training set. For fairness, we accordingly fine-tune our 7B model on the same training set, and the results are reported in Table 14. Notably, both VideoChat-R1 and TimeZero are trained with GRPO-based reinforcement learning, incorporating task-specific temporal rewards to enhance video reasoning. Moreover, their backbone LLMs are based on the stronger Qwen2.5 family, whereas our model relies on the earlier Qwen2 backbone. Despite these advantages, our model achieves comparable or even higher performance across key metrics. This demonstrates the effectiveness of our design and the strong data efficiency of TempGCap.

Table 15: Comparison with Attention-based Filtering Methods on NExT-QA.

| Method | Acc (%) |
|---|---|
| MIST (Gao et al., 2023) | 57.2 |
| STR (Li et al., 2023b) | 70.0 |
| GCG (Wang et al., 2024a) | 74.6 |
| Ours | **79.4** |

**Comparison with Attention-based Filtering Methods.** As shown in Table 15, we compare our approach with three representative attention-based filtering methods on NExT-QA: MIST (Gao et al., 2023), STR (Li et al., 2023b), and GCG (Wang et al., 2024a). Among them, MIST achieves an accuracy of 57.2%, STR improves the performance to 70.0%, and GCG further increases it to 74.6%. In contrast, our method attains the highest accuracy of **79.4%**, outperforming GCG by **4.8** points. These results highlight the effectiveness of our method.

**Comparison with Frame Sampling and Token Pruning methods.** As shown in Table 16, we first compare our method with the frame sampling approach AKS (Tang et al., 2025). Although AKS reduces the number of processed frames, its frame selection is performed *before* feeding frames into the LLM and is therefore not guided by the language query within LLM. This unguided strategy often fails to preserve query-relevant visual content and can lead to noticeable information loss. Consequently, AKS achieves only 46.91 R1@0.5 and 43.52 mIoU on Charades-STA, and 26.87 mIoU and 16.57 Acc@IoU on ReXTime. In contrast, our method attains 51.37 R1@0.5 and 47.33 mIoU on Charades-STA, as well as 29.71 mIoU and 18.57 Acc@IoU on ReXTime. These improvements demonstrate that disentangled temporal modeling and LLM-internal guided keyframe selection provide more reliable temporal localization than pre-LLM sampling strategies. We further compare our approach with two token pruning methods, ToMe (Bolya et al., 2022) and AIM (Zhong et al., 2025). While these methods effectively reduce the number of visual tokens passed to the LLM, they cannot guarantee the preservation of essential spatio-temporal cues. As a result, ToMe yields only 40.13 R1@0.5 and 39.31 mIoU on Charades-STA, and 25.15 mIoU and 14.55 Acc@IoU

Table 16: Comparison with Frame Sampling and Token Pruning methods.

| Method | LLM TFLOPs | Charades-STA | | ReXTime | |
|---|---|---|---|---|---|
| | | R1@0.5 | mIoU | mIoU | Acc@IoU |
| AKS (Tang et al., 2025) | 7.9 | 46.91 | 43.52 | 26.87 | 16.57 |
| ToMe (Bolya et al., 2022) | 10.2 | 40.13 | 39.31 | 25.15 | 14.55 |
| AIM (Zhong et al., 2025) | 9.9 | 49.29 | 45.61 | 27.84 | 17.02 |
| **Ours** | 10.5 | **51.37** | **47.33** | **29.71** | **18.57** |

on ReXTime. AIM improves the results to 49.29 R1@0.5, 45.61 mIoU on Charades-STA and 27.84 mIoU, 17.02 Acc@IoU on ReXTime. Despite operating under similar or even tighter token budgets, our method consistently achieves the highest scores across all benchmarks, highlighting that global token pruning is insufficient for long-form temporal reasoning. Instead, performing query-aware temporal guidance and spatial selection within the LLM preserves critical information while maintaining efficiency.

Table 17: Comparison on Computation and Latency.

| Method | TFLOPs | Latency |
|---|---|---|
| Full | 28.2 | 1642ms |
| Slow-Fast | 16.2 | 1311ms |
| Ours | 10.5 | 1397ms |

**Computation and Latency Analysis.** Divid achieves a substantial reduction in computational cost while maintaining competitive inference speed. Table 17 summarizes the comparison. Before engineering optimization, the total latency of our method is 1530 ms, which is slightly higher than the Slow-Fast variant at 1311 ms but remains lower than the full model at 1642 ms. After applying several engineering optimizations, such as parallelizing computations during keyframe selection, the latency is reduced from 1530 ms to 1397 ms. This improvement narrows the gap between the theoretical reduction in FLOPs and the practical inference speed. All latency measurements were obtained under the same conditions on an NVIDIA A100 GPU. Divid requires only 10.5 TFLOPs, corresponding to a 63 percent reduction compared with the full model that requires 28.2 TFLOPs and a 35.2 percent reduction compared with the Slow-Fast baseline. Despite reducing FLOPs by more than one third relative to Slow-Fast, Divid increases latency by only 6.5 percent and achieves consistently higher performance across all evaluated metrics. These results demonstrate that Divid provides an effective balance between efficiency and accuracy.

**Visualization of token-wise fusion weights produced by the spatio-temporal soft-router.** To provide a deeper understanding of how the proposed soft-router modulates the contributions of temporal and spatial information at the level of individual query tokens, we conduct a comprehensive visualization and analysis based on aggregated routing behavior across all decoder layers. For every query token in the input sequence, we extract the temporal and spatial fusion weights produced at each decoder layer. These per-layer weights are then averaged to obtain a stable and layer-agnostic fusion score that reflects the final routing behavior of the model. This procedure yields two scalar quantities per token, corresponding respectively to its temporal preference and spatial preference. After computing these scores for the entire vocabulary of query tokens encountered during evaluation on ReXTime benchmark, we sort the tokens according to their temporal and spatial fusion weights and present the Top-100 tokens for each category in Figure 3. This ranking-based approach highlights the tokens for which the routing mechanism expresses the clearest preference, facilitating more interpretable inspection of the model's internal fusion dynamics. The results reveal consistent and semantically meaningful routing patterns. Tokens with explicit or implicit temporal semantics, including "then", "when", and "start", exhibit markedly higher temporal fusion weights. These tokens frequently appear in contexts that require reasoning about temporal ordering, event progression, or causal relationships, and the router accordingly prioritizes temporal features when processing them. In contrast, tokens that denote concrete objects, visual attributes, or detailed physical interactions are associated with significantly higher spatial fusion weights. Representative examples include nouns such as "lady" and "dog", as well as action verbs such as "applying" and "holding". These tokens intrinsically depend on fine-grained object appearance or spatial configuration, and the router therefore amplifies the influence of spatial representations to support more precise visual grounding. This visualization provides strong evidence that the soft-router does not operate through uniform or

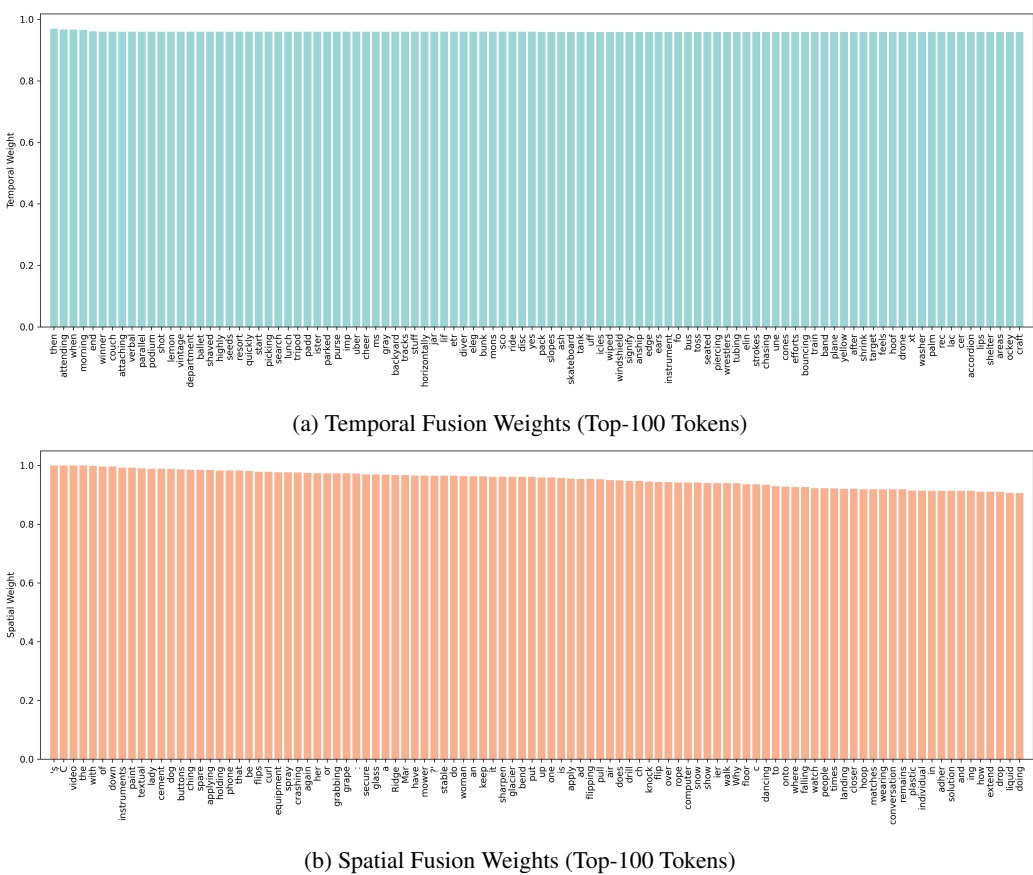

(a) Temporal Fusion Weights (Top-100 Tokens)

(b) Spatial Fusion Weights (Top-100 Tokens)

Figure 3: Visualization of token–wise fusion weights produced by the spatio-temporal soft-router.

indiscriminate weighting. Instead, it systematically adapts the degree of temporal and spatial fusion based on the semantic role of each token, producing token-dependent fusion strategies that are aligned with the underlying linguistic and visual demands. Such behavior supports the objective of disentangled spatial–temporal reasoning within the decoder and demonstrates that the router learns structured decision patterns that enhance both interpretability and task-specific temporal grounding performance.

**Visualization of failure cases.** As shown in Figure 4, we visualize two representative failure cases from the ReXTime benchmark and compare our Divid framework with the full-attention baseline. In both examples, two models correctly answer the multiple-choice question, yet their predicted temporal intervals obtain very low tIoU scores (below 0.5). This highlights a common challenge in grounded VideoQA: producing the correct answer does not necessarily imply correct temporal grounding. In the upper example, the question asks for the event after the "CrossFit Connect" logo appears. The baseline incorrectly grounds only the moment when the logo is shown, whereas the ground truth corresponds to the subsequent content. Our model predicts a longer segment that covers both the query cue and the answer-relevant interval, resulting in a higher tIoU, but the overall localization remains imperfect. In the lower example, the question asks for the action the man performs before he drops the weights on the floor. The baseline prediction localizes the dropping motion itself, which corresponds only to the question cue rather than the answer. Our model instead produces a segment that successfully includes the action occurring prior to the drop, leading to a more aligned but still incomplete interval. Such errors likely arise because most existing training samples place the question and the answer within the same temporal region, providing limited supervision for queries that require grounding outside the question segment (e.g., "before" or "after"). Nevertheless, our model consistently shows better coverage of answer-relevant intervals than the full-attention baseline. In future work, we plan to incorporate more training data with non-overlapping question-answer intervals and improve the model's ability to perform cross-interval temporal grounding.

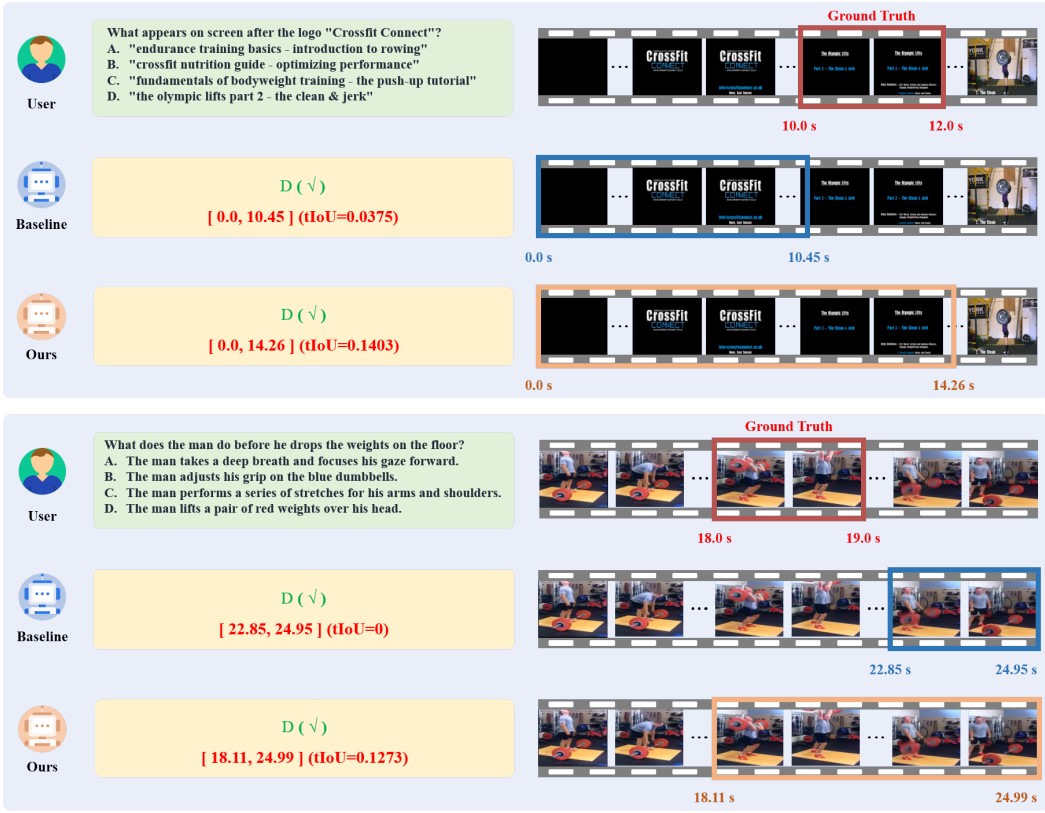

Figure 4: Visualization of failure cases on ReXTime benchmark.

**Visualization of Key Frame Selection.** Figure 5 visualizes the key-frame selection behavior across all decoder layers. Dark blue cells indicate frames selected as key frames, while light blue cells denote non-selected frames. We observe that a small portion of selected frames falls outside the ground-truth segment; while this is not ideal, these frames also contain work-related activities, which may attract attention given the query. Nevertheless, the vast majority of key frames concentrate on the target segment, aligning well with our predicted temporal window. This demonstrates that the temporal branch effectively highlights query-relevant frames and guides the spatial branch toward the correct video region.

## B    DETAILS OF TEMPGCAP DATASET

Table 18: Comparison of instruction-tuning datasets for temporally grounded video understanding. "M" denotes manual annotation, "A" denotes automatic annotation.

| Dataset | QA-pairs | #Source Videos | Total Duration | Avg. Duration | Annotation |
|---|---|---|---|---|---|
| VTimeLLM-Stage2 (Huang et al., 2024a) | 146K | 146K | 2,124h | 52s | A |
| InternVid-G (Wang et al., 2024e) | 715K | 83K | 10,106h | 438s | A |
| Momentor (Qian et al., 2024a) | 10.4M | 65K | 7,260h | 403s | A |
| Ours | 559K | 428K | 8,831h | 73s | M + A |

We present **TempGCap**, a large-scale instruction tuning dataset for *timestamp-guided captioning*, where models are required to generate natural language descriptions for specific temporal segments within untrimmed videos. To construct TempGCap, we aggregate timestamp-aligned video-text pairs from diverse sources and process them through a unified three-stage pipeline. The final dataset contains 559K samples from 428K videos, totaling over 8K hours of content. As shown in Table 18, compared with existing instruction-tuning datasets such as VTimeLLM-Stage2 (Huang et al., 2024a), InternVid-G (Wang et al., 2024e), and Momentor (Qian et al., 2024a), TempGCap achieves a better balance between scale, diversity, and annotation quality, benefiting from both manual and

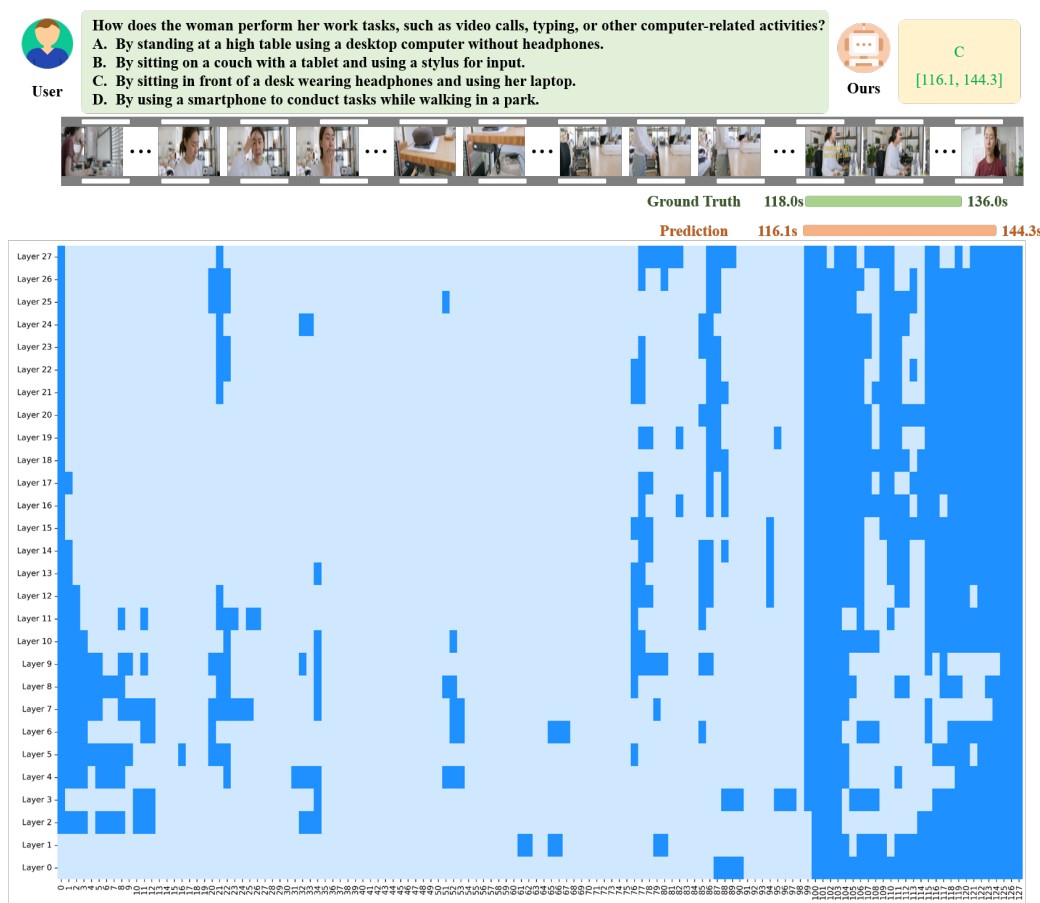

Figure 5: Visualization of Key Frame Selection.

automatic supervision. This unique combination provides higher temporal precision and broader coverage across video domains, making it particularly suitable for training models on temporally grounded video understanding tasks.

## B.1 DATA COLLECTION AND PROCESSING

The **TempGCap** dataset is built by aggregating timestamp-aligned video-text pairs from a broad range of publicly available and temporally rich video-language datasets, covering diverse domains such as everyday activities, instructional content, and movie recaps. Representative sources include DiDeMo (Anne Hendricks et al., 2017), ActivityNet Captions (Krishna et al., 2017), HACS (Zhao et al., 2019), and Tarsier2-Recap-585K (Yuan et al., 2025). To construct high-quality timestamp-guided captioning samples from these heterogeneous sources, we design a three-stage data processing pipeline tailored to the characteristics and limitations of each dataset.

**Sub1: Reannotating Untrimmed Videos with Manual Temporal Annotations.** We begin with datasets that already provide temporal grounding annotations or segment-level boundaries. For the grounding datasets DiDeMo (Anne Hendricks et al., 2017) and ActivityNet Captions (Krishna et al., 2017), we directly adapt their manually annotated temporal segments into the timestamp-guided captioning format. For the temporal action localization dataset HACS (Zhao et al., 2019), which includes human-annotated temporal boundaries but lacks natural language descriptions, we generate detailed captions for each action segment using the advanced video captioning model Tarsier (Wang et al., 2024c). To ensure the quality of the generated captions, we perform a two-stage filtering process combining rule-based checks and LLM-based scoring. First, we remove overly long captions, captions containing excessive shot-change descriptions (detected via keyword matching), and captions dominated by static scene descriptions. To quantify static-content dominance, we use Qwen2-7B (Yang et al., 2024) to assign a static-content score from 0 to 5 and discard captions with a score

higher than 3. Second, we conduct random human audits to further verify the alignment and descriptive accuracy of the remaining captions. Since Tarsier tends to produce relatively detailed and lengthy descriptions, we adapt the instruction format in our training samples by using prompts of the form "Describe ... in detail," as illustrated in Figure 6 (b). This ensures consistency between the captioning style of the data and the model's instruction-following behavior.

To support the construction of **Sub2** and **Sub3**, we introduce the Tarsier2-Recap-585K dataset (Yuan et al., 2025), a large-scale, high-quality corpus of short video clips. It comprises 585K distinct clips spanning 1,972 hours, sourced from open-domain datasets. Each clip (5–20 seconds) is annotated with a detailed caption generated by Tarsier2-7B (Yuan et al., 2025), which surpasses GPT-4o in generating accurate and descriptive captions for short video segments. To prevent data leakage during zero-shot evaluation and streamline data handling, we filter out subsets such as Charades, Charades-Ego (Sigurdsson et al., 2018), Ego4D (Grauman et al., 2022), and TGIF (Li et al., 2016). The retained subsets, including ActivityNet (Krishna et al., 2017), Kinetics-700 (Carreira & Zisserman, 2017), VATEX (Wang et al., 2019), LSMDC (Rohrbach et al., 2017), Oops (Epstein et al., 2020), SomethingSomethingV2 (Goyal et al., 2017), TREC-VTT (Awad et al., 2023) and WebVid (Bain et al., 2021), cover a broad spectrum of video domains and provide diverse supervision for timestamp-guided captioning.

**Sub2: Recovering Untrimmed Contexts from Captioned Clips.** For several retained subsets such as ActivityNet, Kinetics-700, and VATEX, the original untrimmed videos are publicly accessible. We retrieve these full-length videos and reconstruct timestamp-guided captioning samples by aligning each captioned clip with its corresponding original timestamps. Importantly, the caption is kept unchanged and strictly corresponds to the original trimmed segment; the additional untrimmed context functions solely as surrounding background, encouraging the model to distinguish relevant content from temporally adjacent but irrelevant regions. To further refine segment boundaries and improve temporal precision, we adopt a feature-based boundary adjustment procedure. For each boundary frame, we extract ViT embeddings of the frame and its adjacent neighbor and compute cosine similarity. If the similarity falls below a threshold of 0.9, we iteratively expand the start or end timestamp outward along the temporal axis until the similarity stabilizes. The maximum expansion range is capped (e.g., 1.5 seconds) to avoid excessive drift, and segments whose boundary neighborhoods exhibit consistently low similarity across multiple consecutive frames are discarded. This refinement removes clips with ambiguous transitions and yields more reliable temporal anchors. We additionally apply several quality-control filters to the recovered untrimmed videos, including discarding corrupted downloads, removing videos with abnormal durations (e.g., shorter than 5 seconds), and eliminating duplicates. Finally, we perform random human audits to ensure that the refined segments remain semantically aligned with their captions and that the restored untrimmed contexts do not introduce misleading artifacts.

**Sub3: Synthesizing Pseudo-Untrimmed Videos.** For subsets where untrimmed videos are unavailable, we simulate untrimmed contexts by synthesizing pseudo-untrimmed sequences. Concretely, for each target clip, we randomly sample 2 to 4 additional clips and concatenate them in a shuffled order to form a longer video sequence. We also control the distribution of the inserted segments to ensure that target clips appear at different positions within the synthesized video. The timestamp of the target clip is recalculated based on its position in the sequence, and its original caption is retained. This synthesis strategy enables us to construct timestamp-guided captioning instances that emulate real-world untrimmed video scenarios with surrounding contextual distractions, helping the model learn to perform temporal grounding under weak and noisy supervision. To avoid distractor clips that are too similar to the target segment, we perform filtering at both textual and visual levels. For textual similarity, we use a BERT (Devlin et al., 2019) model to compute caption similarity between the target clip and each candidate distractor clip, and discard distractors with similarity above 0.5. For visual similarity, we use a UMT (Li et al., 2023a) model to extract features from 8 sampled frames of each clip and remove distractors whose similarity exceeds 0.4. Clips whose captions have been repeatedly used are also filtered out to prevent caption duplication. Finally, we perform manual spot-checking to ensure that the synthesized pseudo-untrimmed sequences are reasonable and that distracting segments do not unintentionally overlap in semantics with the target clip.

**Data and Domain Leakage Prevention.** To prevent data leakage and ensure that our instruction-tuning corpus does not overlap with downstream evaluation benchmarks, we adopt strict filtering at the video identity, subset, and domain levels. First, we remove any video whose *name-based identifier* (e.g., `youtube_id` or other dataset-specific video IDs) exactly matches those used in our

Table 19: TempGCap dataset statistics grouped by annotation strategy and domain.

| Name | Domain Category | #Videos | #Samples | Avg. Duration | Duration | Caption Len. (w) |
|---|---|---|---|---|---|---|
| *Sub1: Reannotating Untrimmed Videos with Manual Temporal Annotations* | | | | | | |
| DiDeMo | Open Domain | 7,678 | 30,158 | 28.2 s | 60.2 h | 7.5 |
| ActivityNet Captions | Human Activities / Events | 10,009 | 37,421 | 117.2 s | 325.8 h | 13.5 |
| HACS | Human Activities | 21,552 | 68,705 | 139.3 s | 834.1 h | 86.3 |
| *Sub1 Sum.* | - | 39,239 | 136,284 | 111.9 s | 1220.1 h | 48.9 |
| *Sub2: Recovering Untrimmed Contexts from Captioned Clips* | | | | | | |
| VATEX | Multilingual & Crowd Scenes | 22,422 | 22,425 | 165.7 s | 1031.9 h | 82.1 |
| ActivityNet | Human Activities | 9,075 | 32,595 | 115.7 s | 291.6 h | 97.8 |
| Kinetics-700 | Human Actions | 49,660 | 49,673 | 149.6 s | 2063.9 h | 81.5 |
| *Sub2 Sum* | - | 81,157 | 104,693 | 150.3 s | 3387.4 h | 86.7 |
| *Sub3: Synthesized from Short Clips without Untrimmed Videos* | | | | | | |
| Oops | Accident / Unexpected Events | 7,948 | 7,948 | 39.1 s | 86.3 h | 80.7 |
| SomethingSomethingV2 | Object-centric Human Actions | 9,996 | 9,996 | 15.0 s | 41.8 h | 67.5 |
| TREC-VTT | Web Video / Diverse Topics | 14,199 | 14,199 | 25.2 s | 99.4 h | 81.0 |
| LSMDC | Movie / Narrative Video | 108,271 | 108,271 | 16.4 s | 492.8 h | 72.8 |
| WebVid | Web-scale Open-domain | 177,909 | 177,909 | 71.0 s | 3503.6 h | 71.3 |
| *Sub3 Sum* | - | 318,323 | 318,323 | 47.8 s | 4223.9 h | 72.3 |

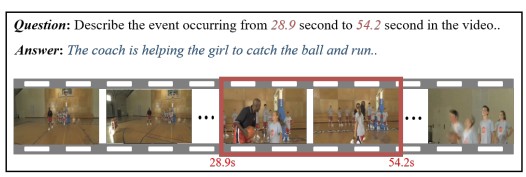 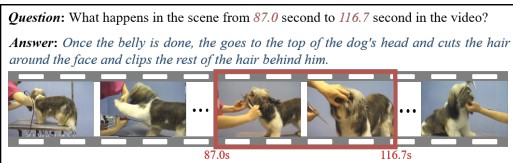

(a) Samples converted from dense video caption datasets

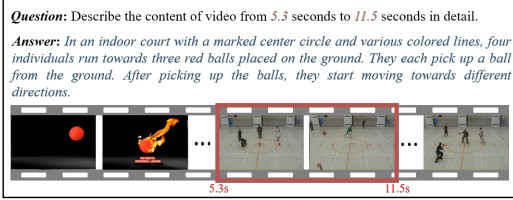 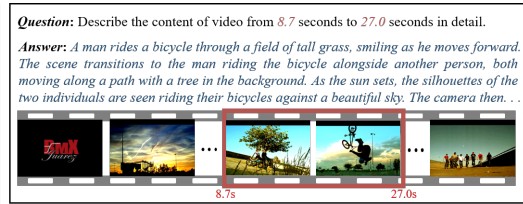

(b) Samples annotated by Tarsier

Figure 6: Visualization of Training samples.

evaluation benchmarks, ensuring that no training sample contains content from test videos. Second, to avoid domain-level leakage, we manually exclude video sources that closely resemble the target benchmarks. In particular, we filter out datasets such as Charades, Charades Ego (Sigurdsson et al., 2018), Ego4D (Grauman et al., 2022), and TGIF (Li et al., 2016), whose content or annotation style strongly overlaps with temporal grounding and VideoQA benchmarks. Third, for open-domain datasets that contain officially defined training, validation, and test partitions, such as DiDeMo (Anne Hendricks et al., 2017), HACS (Zhao et al., 2019), ActivityNet Captions (Krishna et al., 2017), and Kinetics 700 (Carreira & Zisserman, 2017), we explicitly retain only the training split and manually exclude all validation and test subset videos to avoid any accidental overlap with downstream evaluation. This three-stage filtering process ensures that TempGCap is clean, non-overlapping, and free of unintentional data or domain leakage under the zero-shot evaluation setting.

## B.2 DATASET STATISTICS

Table 19 presents detailed statistics of the TempGCap dataset, categorized according to the underlying annotation strategy. The dataset comprises a total of 559,300 timestamp-guided captioning samples derived from untrimmed videos, with a cumulative duration of approximately 8,831.4 hours.

**Sub1** includes 136,284 samples from 39,239 videos, totaling 1,220.1 hours. The average clip duration is 111.9 seconds, and the average caption length is 48.9 words.

**Sub2** includes 104,693 samples from 81,157 videos, totaling 3,387.4 hours. The average clip duration is 150.3 seconds, and the average caption length is 86.7 words.

**Sub3** includes 318,323 samples from 318,323 videos, totaling 4,223.9 hours. The average clip duration is 47.8 seconds, and the average caption length is 72.3 words.

Some video sources appear in multiple subsets. After deduplication, the dataset contains approximately **428K** unique videos.

### B.3 VISUALIZATION

To better illustrate the characteristics of the captions used in TempGCap and to analyze the potential noise introduced by the semi-automated annotation process, we provide qualitative comparisons in Figure 6. The upper examples correspond to samples converted from existing dense-caption datasets, where the descriptions typically summarize coarse, event-level content and often overlook fine-grained or temporally localized actions. In contrast, the lower examples show captions generated by Tarsier, which contain richer action-centric details, clearer temporal progression, and more explicit object–action interactions. Although such fine-grained annotations differ from the style of earlier dense-caption datasets, this shift is not detrimental; on the contrary, the increased granularity provides stronger supervision for fine-grained temporal reasoning. Furthermore, by consistently using prompts of the form "Describe ... in detail" during training, the model is explicitly guided to differentiate between coarse-grained and fine-grained description styles. This reduces the potential impact of stylistic inconsistencies across data sources and allows the model to benefit from the additional temporal specificity provided by Tarsier-generated captions.

## C DETAILS OF TRAINING

**Training Loss.** All tasks in our framework, including timestamp-guided captioning, temporal grounding, and grounded VideoQA, are formulated as conditional text generation. Following standard practice in large language model (LLM) training, we adopt the conventional autoregressive language modeling objective without introducing task-specific heads or losses.

For each training example indexed by $n \in \{1, \ldots, N\}$, let $x^{(n)}$ denote the multimodal input (e.g., video frames and textual prompt), and let

$$y^{(n)} = \{y_1^{(n)}, \ldots, y_{T^{(n)}}^{(n)}\}$$

denote its target output token sequence of length $T^{(n)}$. The decoder models the standard autoregressive probability

$$p_\theta\left(y_t^{(n)} \,\middle|\, y_{<t}^{(n)}, x^{(n)}\right), \tag{8}$$

where $y_{<t}^{(n)} = \{y_1^{(n)}, \ldots, y_{t-1}^{(n)}\}$ and $\theta$ denotes model parameters.

During training, we apply teacher forcing and compute the next-token prediction loss. Let $\mathcal{T}^{(n)} \subseteq \{1, \ldots, T^{(n)}\}$ denote the index set of *prediction tokens*, including all answer tokens as well as timestamp tokens. Following common LLM practice, we average the negative log-likelihood over *all* prediction tokens in the batch:

$$\mathcal{L} = -\frac{1}{\sum_{n=1}^{N} |\mathcal{T}^{(n)}|} \sum_{n=1}^{N} \sum_{t \in \mathcal{T}^{(n)}} \log p_\theta\left(y_t^{(n)} \,\middle|\, y_{<t}^{(n)}, x^{(n)}\right). \tag{9}$$

This token-wise averaged language modeling loss is used for all tasks. Temporal boundaries are represented directly as text tokens, so no additional regression objectives, bounding-box losses, or threshold heuristics are required. Similarly, we do not introduce auxiliary losses or class-balancing terms; any imbalance across datasets is handled purely through data sampling strategies (e.g., up-sampling timestamp-grounded examples), rather than modifying the optimization objective.

**Training Setting.** Table 20 summarizes the training configurations used in Stage I and Stage II. Stage I adopts a temporal input of 32 frames with 2 keyframes and a global batch size of 256. Stage II uses 128 frames with 32 keyframes and a global batch size of 128. The model is trained

Table 20: Training configuration for Stage I and Stage II.

| Config | Stage I | Stage II |
|---|---|---|
| LLM backbone | Qwen2-7B | |
| ViT backbone | ViT-g/14 | |
| Frame resolution | $224 \times 224$ | |
| Temporal resolution ($T' \times H' \times W'$) | $32 \times 4 \times 4$ | $128 \times 4 \times 4$ |
| Spatial resolution ($T \times H \times W$) | $32 \times 8 \times 8$ | $128 \times 8 \times 8$ |
| Number of key frames $k$ | 2 | 32 |
| Global batch size | 256 | 128 |
| Batch size per GPU | 2 | 1 |
| Accumulated steps | 4 | 4 |
| DeepSpeed zero stage | 2 | 2 |
| Learning rate | $1 \times 10^{-3}$ | $2 \times 10^{-5}$ |
| Learning rate schedule | cosine decay | |
| Warmup ratio | 0.03 | |
| Weight decay | 0 | |
| Epoch | 1 | |
| Optimizer | AdamW | |
| Precision | bf16 | |
| Max length | 8192 | |

using the AdamW optimizer with a cosine decay learning rate schedule and a warmup ratio of 0.03. Both stages apply DeepSpeed ZeRO Stage 2 and bf16 precision to improve memory efficiency. The maximum token length is set to 8192, and gradients are accumulated over 4 steps per GPU. For better multimodal understanding, we initialize the LLM weights with those from Qwen2-VL (Wang et al., 2024d), while adopting ViT-g (Sun et al., 2023) as the vision encoder. To enhance temporal grounding capability, we perform resampling of grounding data from DiDeMo (Anne Hendricks et al., 2017) and ActivityNet Captions (Krishna et al., 2017), allowing the model to encounter each sample three times during training. For DeVE-QA (Qin et al., 2025), in addition to using it for question-answering training, we also repurpose its questions as queries to construct grounding tasks. This design enables the model to better adapt to question-form queries in grounding scenarios. The experiments are conducted on 32 A100 GPUs. Stage I training takes approximately 3 hours, and Stage II takes around 80 hours.

## D  BENCHMARKS

We assess the performance of Divid across multiple benchmarks that target temporal grounding and grounded video question answering (VideoQA). The core datasets used in our evaluation are summarized as follows.

Charades-STA (Gao et al., 2017) consists of approximately 10,000 indoor videos, each averaging around 30 seconds. It provides over 16,000 temporal annotations aligned with natural language descriptions of everyday activities. This dataset is widely adopted for evaluating temporal localization capabilities in constrained indoor scenarios.

CG-Bench (Chen et al., 2025) focuses on grounded VideoQA in extended videos. It comprises 1,200 curated videos spanning 10 to 80 minutes across diverse topics, accompanied by 12,000 QA pairs. The benchmark classifies questions into three categories—perception, reasoning, and hallucination—and introduces evaluation paradigms based on "white box" and "black box" clue verification, aiming to measure the fidelity of model predictions with respect to visual evidence.

NExT-GQA (Xiao et al., 2024), building on the NExT-QA dataset, emphasizes reasoning over causal and temporal events. It offers 10,500 QA samples with annotated temporal spans, focusing on questions that require causal inference and temporal sequencing. The original descriptive question type from NExT-QA is excluded to better target event understanding over time.

ReXTime (Chen et al., 2024a) presents a benchmark for high-level temporal reasoning in videos, using a semi-automated pipeline to generate QA pairs while ensuring annotation accuracy through manual verification. The dataset includes 921 validation and 2,100 test instances. Notably, there exists a substantial 14.3% performance gap between state-of-the-art models and human annotators, highlighting the challenge of understanding causal structures in complex temporal narratives.

## ETHICS STATEMENT

All datasets used are publicly available and commonly adopted in prior research, and no private or sensitive data is included. Our model is intended for academic research only, and we caution against misuse in applications such as surveillance or disinformation.

## REPRODUCIBILITY STATEMENT

We provide detailed descriptions of our model architecture (Section 3), dataset construction (Section 3.3 and Appendix B), training setups including hyperparameters and hardware (Section 4.1 and Appendix C), and evaluation protocols for all benchmarks (Section 4.2 and Appendix D). Ablation studies (Section 4.4 and Appendix A) further validate individual design choices. We will release the trained models and the TempGCap dataset to facilitate reproducibility and future research.

## USE OF LLMS

We acknowledge that a large language model (LLM) is used during the preparation of this manuscript for the purpose of grammar checking and minor language refinement only. The LLM does not contribute to the core methodology, experiments, analyses, or results. All scientific content, claims, and conclusions are the sole responsibility of the authors.

