# OpenReview forum: "Divid: Disentangled Spatial-Temporal Modeling within LLMs for Temporally Grounded Video Understanding"
_ICLR.cc/2026/Conference — ICLR 2026 Poster_

### Official Review · Reviewer_mbkF · 2025-10-15

**Soundness:** 3
**Presentation:** 3
**Contribution:** 3
**Rating:** 6
**Confidence:** 4

**Summary:**

This paper introduces Divid, a dual-branch framework for video understanding that explicitly disentangles spatial and temporal modeling within the Large Language Model (LLM) decoder to improve temporally grounded tasks in long-form videos. The architecture features a temporal branch that processes dense, low-resolution frames to capture global motion dynamics, which in turn guides a spatial branch in selecting a sparse set of high-resolution keyframes relevant to the query. A lightweight spatio-temporal soft-router is proposed to adaptively fuse the information from these two branches at the token level for the final prediction. To support this model, the authors also constructed a large-scale dataset, TempGCap, containing 559K timestamp-grounded video-text pairs to provide rich temporal supervision. Experiments conducted on several temporal grounding and grounded VideoQA benchmarks show the effectiveness of Divid compared to existing methods.

**Strengths:**

* The core contribution is the dual-branch structure that disentangles spatio-temporal information within the LLM decoder. The idea of using dense, low-resolution frames for temporal modeling to guide the selection of sparse, high-resolution keyframes for spatial modeling is intuitive and sound. Combined with the token-level soft-router for adaptive fusion, this approach effectively addresses the challenges of high computational cost and spatio-temporal entanglement in long video processing.

* The paper identifies the shortcomings of existing instruction-tuning datasets in temporal precision and diversity and builds a high-quality dataset of 559K samples using three complementary strategies (manual annotation, recovering untrimmed contexts, and synthesizing pseudo-long videos). This will be highly valuable for future research in the community.

**Weaknesses:**

* Although the appendix shows a significant reduction in computational cost (TFLOPs), the reported latency is slightly higher than the slow-fast baseline. The authors acknowledge this is due to the extra keyframe selection step and suggest it can be improved with engineering optimizations, but it remains a minor weakness where the theoretical efficiency gains do not fully translate to practical inference speedup.

* The construction of the TempGCap dataset partially relies on other models (e.g., using the Tarsier model to generate captions), which could introduce model-specific biases or errors into the dataset. While this is a practical approach for large-scale data creation, a more in-depth analysis of the potential noise introduced by this semi-automated process would have further strengthened the claims about dataset quality.

*  The core temporal-guided keyframe selection mechanism (Temporal Attention Scores -> Top-K Selection) is conceptually similar to other attention-based filtering methods [1, 2, 3]. The paper could better contextualize its approach by comparing it more directly with prior work in frame sampling or token pruning to highlight its uniqueness in the context of Video LLMs.

[1] MIST: Multi-modal Iterative Spatial-Temporal Transformer for Long-form Video Question Answering

[2] Discovering Spatio-Temporal Rationales for Video Question Answering

[3] Weakly Supervised Gaussian Contrastive Grounding with Large Multimodal Models for Video Question Answering

**Questions:**

Please see Weaknesses

---

> ### Author Response · Authors · 2025-11-25
>
> We sincerely appreciate your constructive feedback and your positive assessment of our contributions. We are glad that you find the Divid framework and the TempGCap dataset valuable for temporally grounded video understanding. We have carefully addressed all your questions and suggestions in the detailed response below and have incorporated all corresponding revisions and additional experimental results in the updated manuscript (highlighted in blue). We hope that these improvements help clarify the strengths and significance of our work and that they may positively influence your overall evaluation.
>
> >[**Q1**]. Although the appendix shows a significant reduction in computational cost (TFLOPs), the reported latency is slightly higher than the slow-fast baseline. The authors acknowledge this is due to the extra keyframe selection step and suggest it can be improved with engineering optimizations, but it remains a minor weakness where the theoretical efficiency gains do not fully translate to practical inference speedup.
>
> [**A1**]. We thank the reviewer for the valuable comment. As shown in Table R6, after applying several engineering optimizations such as parallelizing computations during keyframe selection, the latency of our system is reduced from 1530 ms to 1397 ms. This improvement narrows the gap between the theoretical reduction in FLOPs and the practical inference speed. Importantly, compared with the Slow-Fast baseline, Divid reduces FLOPs by 35.2 percent while increasing latency by only 6.5 percent, and it achieves higher performance across all evaluation metrics. These results indicate that the small latency overhead does not affect the overall efficiency advantage or the practical applicability of our framework.
>
> #### Table R6. Computation and Latency Analysis.
>
> | Method    | TFLOPs | Latency |
> |-----------|--------|---------|
> | Full      | 28.2   | 1642ms  |
> | Slow-Fast | 16.2   | 1311ms  |
> | Ours      | 10.5   | 1530ms  |
> | *Ours     | 10.5   | 1397ms  |

---

> ### Author Response · Authors · 2025-11-25
>
> >[**Q2**]. The construction of the TempGCap dataset partially relies on other models (e.g., using the Tarsier model to generate captions), which could introduce model-specific biases or errors into the dataset. While this is a practical approach for large-scale data creation, a more in-depth analysis of the potential noise introduced by this semi-automated process would have further strengthened the claims about dataset quality.
>
> [**A2**]. Thank you for the reviewer’s thoughtful comment. We agree that relying on model-generated captions may introduce model-specific biases or noise. To address this concern, we have expanded our explanation of the caption filtering and quality-control pipeline and added a new visualization analysis in Figure 6 to examine the characteristics of Tarsier-generated captions.
>
> First, we provide a more detailed description of our caption quality-control process in Sub1. For datasets such as DiDeMo and ActivityNet Captions, we directly use their human-annotated temporal segments. For HACS, although temporal boundaries are manually labeled, natural-language descriptions are not provided; therefore, we generate fine-grained captions using the Tarsier model. To mitigate potential noise introduced by this semi-automated annotation, we adopt a two-stage filtering strategy. In the first stage, we apply rule-based checks to remove captions that are excessively long, dominated by shot-change descriptions, or contain an overly large amount of static content. In the second stage, we compute static-content scores using Qwen2-7B and discard captions with high scores. We also conduct random human audits to verify temporal alignment and ensure descriptive accuracy. Since Tarsier tends to produce detailed descriptions, we unify the dataset’s instruction style by using prompts such as “Describe in detail,” which helps maintain stylistic consistency during training.
>
> Second, to assess whether Tarsier introduces undesirable biases, we include a visualization analysis in Figure 6. The comparison shows that captions converted from dense-caption datasets are often coarse and event-level, whereas Tarsier-generated captions provide richer action-centric details, clearer temporal progression, and more explicit object–action interactions. These differences reflect a shift toward finer descriptive granularity rather than harmful noise. Such granularity is beneficial for timestamp-guided video understanding because it strengthens the model’s ability to perform fine-grained temporal reasoning. Moreover, consistently prompting Tarsier with “Describe in detail” encourages the model to differentiate between coarse and fine-grained caption styles, thereby reducing stylistic inconsistencies across data sources.
>
> Finally, we provide empirical evidence that TempGCap delivers stronger temporal supervision compared to fully automated datasets. Although TempGCap is much smaller than Momentor-10M (559K versus 10.4M samples), it achieves higher performance on Charades-STA and ReXTime, as shown in Table 8. These results suggest that Momentor’s fully automated pipeline accumulates noise from event segmentation, tracking, and LLM-based caption generation, whereas TempGCap’s hybrid design yields more stable and accurate temporal annotations.

---

> ### Author Response · Authors · 2025-11-25
>
> >[**Q3**]. The core temporal-guided keyframe selection mechanism (Temporal Attention Scores -> Top-K Selection) is conceptually similar to other attention-based filtering methods [1, 2, 3]. The paper could better contextualize its approach by comparing it more directly with prior work in frame sampling or token pruning to highlight its uniqueness in the context of Video LLMs.
>
> [**A3**]. We thank the reviewer for the helpful suggestion. In response, we have added direct comparisons with representative attention-based filtering methods as well as state-of-the-art frame sampling and token pruning approaches to better contextualize our contribution.
>
> As shown in Table R7, on the NExT-QA benchmark, our method achieves the highest accuracy among all attention-based filtering methods [1–3]. These results indicate that performing keyframe selection inside the LLM using query-conditioned temporal attention provides clearer temporal grounding than filtering approaches that rely solely on visual saliency.
>
> Furthermore, Table R8 presents comparisons with advanced frame sampling (AKS) and token pruning methods (ToMe, AIM) under matched experimental settings. Despite operating under comparable or even lower computational budgets, our method consistently delivers superior performance across Charades-STA and ReXTime. This demonstrates the advantage of combining disentangled temporal modeling with LLM-internal, query-guided keyframe selection, which retains essential temporal cues more effectively than pre-LLM sampling or global token pruning strategies.
>
> Overall, these results collectively verify both the uniqueness and effectiveness of our design in the context of Video LLMs.
>
> #### Table R7. Comparison with attention-based filtering methods [1–3] on NExT-QA.
>
> | Method | Next-QA Acc (%) |
> |--------|-----------------|
> | [1]    | 57.2            |
> | [2]    | 70.0            |
> | [3]    | 74.6            |
> | Ours   | **79.4**        |
>
> #### Table R8. Comparison with frame sampling and token pruning methods.
>
> | Method              | LLM TFLOPs | Charades-STA R1@0.5 | Charades-STA mIoU | ReXTime mIoU | ReXTime Acc@IoU |
> |-------------|------------|----------------------|--------------------|--------------|------------------|
> | AKS [4]   | 7.9          |46.91                  |   43.52                |  26.87           | 16.57               |
> | ToMe [5]   | 10.2          | 40.13                   | 39.31                 | 25.15            | 14.55               |
> | AIM [6]   |   9.9       | 49.29                   | 45.61                 | 27.84            | 17.02               |
> | **Ours**  | 10.5   | **51.37**           | **47.33**       | **29.71**    | **18.57**       |
>
>
>
> [1] MIST: Multi-modal Iterative Spatial-Temporal Transformer for Long-form Video Question Answering
>
> [2] Discovering Spatio-Temporal Rationales for Video Question Answering
>
> [3] Weakly Supervised Gaussian Contrastive Grounding with Large Multimodal Models for Video Question Answering
>
> [4] Adaptive Keyframe Sampling for Long Video Understanding
>
> [5] Token merging: Your vit but faster
>
> [6] AIM: Adaptive Inference of Multi-Modal LLMs via Token Merging and Pruning

---

> > ### Comment · Reviewer_mbkF · 2025-11-26
> >
> > Thanks for the authors' explanations. I have no questions, and I keep my positive assessment for this paper.

---

> ### Author Response · Authors · 2025-11-26
>
> Thank you again for your time and for your positive assessment of our work. We are very glad to hear that our clarifications have fully addressed your concerns. If you feel that the current version **better reflects the contribution and quality** of the paper, we would be sincerely grateful if you could consider **updating the score accordingly**. In any case, we truly appreciate your support and the constructive feedback you have provided throughout the review process.

---

### Official Review · Reviewer_KBdS · 2025-10-20

**Soundness:** 3
**Presentation:** 3
**Contribution:** 2
**Rating:** 2
**Confidence:** 4

**Summary:**

This paper introduces DIVID, a dual-branch architecture for temporally grounded video understanding using video–language models. The framework explicitly disentangles spatial and temporal modeling within the LLM decoder by employing a temporal branch for dense, low-resolution motion dynamics, and a spatial branch for sparse, high-resolution keyframes guided by temporal attention. The two branches are fused with a query-conditioned spatio-temporal soft-router. In support, the work presents TempGCap, a large-scale dataset of 559k timestamp-grounded video-text pairs. Experiments show strong empirical performance and efficiency gains on several temporal grounding and video QA benchmarks.

**Strengths:**

1. The explicit separation of spatial and temporal processing within the LLM decoder is a clear departure from prior art, which typically performs disentanglement only at the feature extraction or encoder stage. This is well articulated in Figure 1, highlighting the dual-branch structure and adaptive soft-router fusion, which advances the clarity and interpretability of model components.

2. The method's use of temporal attention to guide high-res spatial sampling (rather than uniform or query-agnostic strategies) is empirically backed (Table 6, Figure 1) and brings notable computational and performance improvements, confirmed in ablation studies (Table 5).

3. TempGCap is a substantial new resource, constructed via multiple strategies (manual annotation, recovery, synthesis) for more precise temporally grounded supervisory signals (see Figure 2, Tables 13–14 in the appendix). Dataset quality is assessed in Table 8, showing marked value over prior moment-captioning datasets.

**Weaknesses:**

1. Empirical claims about "state-of-the-art" are largely constrained to the benchmarks provided (Charades-STA, ReXTime, CG-Bench, NExT-GQA). Generalization is briefly touched upon (Table 11), but with limited context; in particular, the extent to which DIVID's improvements generalize to non-temporally grounded or more challenging open-ended video reasoning scenarios is only lightly probed and not explored in depth. Moreover, although some key baselines are included, there are missing baseline comparisons to models adopting more advanced spatio-temporal or cross-modal alignment, as well as absent or only shallow error/mislocalization analysis.


2. The design of the dual-branch decoder and the token-level router is presented in a relatively engineering-driven fashion, with little attempt at formalizing why or under what conditions disentangled decoder branches + soft dynamic fusion should yield better temporal localization or reduce hallucination. There is no theoretical analysis of capacity, representation expressivity, or inductive bias gains. The discussion about why fusion at the token level (rather than head-level, block-level, or post-attention fusion) is beneficial is thin and left to ablations.


3. The TempGCap construction process involves multiple automated and manual annotation stages (as in Figure 2), but more detail should be provided about inter-annotator agreement, filtering criteria, and possible biases introduced (e.g., via pseudo-untrimmed synthesis). Even though Table 13 breaks out annotation strategies, the risk of subtle domain leakage or noise propagation is underexplored.


4. Minor Issues in Exposition:
- There are a few places where notation is not adequately defined or is overloaded (see Section 3.2, Page 4-5).
- Typographical issues, e.g., "the the selected spatial features" (Page 5).
- Some results tables (Table 1, Table 2) reproduce highly dense comparisons, but occasionally lack scale clarity (e.g., 1.5B vs 7B), and the presentation would benefit from clearer highlighting to aid comparison.

**Questions:**

1. Can the authors provide an explicit composite training loss equation, including all task-specific loss terms and weighting, as actually used in implementation (e.g., for temporal localization vs. QA)? How are thresholds, negative sampling, and class imbalance handled during optimization?

2. What is the rationale for choosing a token-level softmax on projected text features for branch fusion? Did the authors experiment with deeper gating modules, or alternative fusion placements (e.g., block, head, or output level)?

3. Do the temporal and spatial branches demonstrably specialize in the types of reasoning intended (e.g., is the spatial branch consistently responsible for frame-level details, or do branches collapse under certain queries)? Would the authors consider adding qualitative analyses or illustrative attention maps?

4. Given the mixture of manual, pseudo, and synthesized captions in TempGCap, how is data leakage prevented on evaluation sets, and have any systematic biases been observed during annotation or curation phases?

5. The most important is, I noticed that the comparative methods adopted by the authors lag behind the latest state-of-the-art research. Recently proposed models such as Qwen2.5-VL, VideoChat-R1, and TimeZero (introduced about six months ago) have achieved strong performance across various temporal localization and general video question answering benchmarks. What are the advantages of the authors’ proposed method compared with these approaches?

---

> ### Author Response · Authors · 2025-11-25
>
> We sincerely appreciate your detailed review, the time you invested in evaluating our submission, and the constructive points you raised. We are grateful that you recognized the strengths of our work, including the novelty of disentangling spatial and temporal modeling inside the LLM decoder, the effectiveness of temporally guided keyframe selection, and the value of constructing the TempGCap dataset. At the same time, we take your concerns seriously and have addressed each of them thoroughly in our responses and with substantial revisions and new experiments in the updated manuscript (highlighted in blue). We genuinely hope that our detailed clarifications, additional experiments, and strengthened manuscript will address your concerns and allow you to reassess the contribution and significance of our work. Thank you again for helping us substantially improve the quality of this submission.

---

> ### Author Response · Authors · 2025-11-25
>
> **Our Responses to Paper Weaknesses:**
> >[**Q1**]. Empirical claims about "state-of-the-art" are largely constrained to the benchmarks provided (Charades-STA, ReXTime, CG-Bench, NExT-GQA). Generalization is briefly touched upon (Table 11), but with limited context; in particular, the extent to which Divid's improvements generalize to non-temporally grounded or more challenging open-ended video reasoning scenarios is only lightly probed and not explored in depth. Moreover, although some key baselines are included, there are missing baseline comparisons to models adopting more advanced spatio-temporal or cross-modal alignment, as well as absent or only shallow error/mislocalization analysis.
>
> [**A1**]. Thank you for the reviewer’s constructive comments. We address the concerns from three perspectives: task motivation, baseline coverage, and error analysis.
>
> **Task Motivation:**
> Our work is specifically motivated by advancing temporally grounded video understanding, particularly in long-video settings where precise temporal localization is essential. The four benchmarks used in our paper, Charades-STA, NExT-GQA, ReXTime, and CG-Bench, are the standard evaluation suites for this task domain, and achieving strong performance on them directly aligns with our core objective. In addition to these grounding-focused datasets, we also include evaluations on MSRVTT-QA, MSVD-QA, and ActivityNet-QA, and Divid achieves competitive results compared with recent VideoLLMs. To address the reviewer’s concern about generalization, we further expanded our analysis of these results. Divid is intentionally designed for temporally grounded reasoning rather than short-video appearance recognition. Its dual-branch decoder, query-guided keyframe selection, and token-level spatio-temporal routing specifically target long-range temporal localization. By contrast, MSRVTT-QA, MSVD-QA, and ActivityNet-QA mainly evaluate object presence, scene attributes, and short-range actions, with limited emphasis on multi-second temporal dependencies. As a result, the advantages of our disentangled temporal modeling do not translate into large gains on these appearance-centric benchmarks, a trend also observed in other grounding-oriented VideoLLMs such as LLaVA-ST and TOGA. Moreover, our instruction-tuning dataset TempGCap is constructed to provide timestamp-grounded supervision that benefits temporal grounding but is not directly aligned with the objectives of generic VideoQA tasks. Even so, Divid still delivers competitive performance compared with recent VideoLLMs, confirming that its specialized design for temporally grounded long-video reasoning does not compromise broader video understanding capability.
>
> **Baseline Coverage:**
> Regarding the reviewer’s concern about missing baseline comparisons, our original submission already includes strong and widely used grounding-oriented models such as Qwen2.5-VL, VideoChat-Flash, TimeSearch, and VideoMind, and Divid already achieves the best performance among these baselines. To further reinforce the comprehensiveness of our evaluation, we have additionally incorporated fair comparisons with the latest advanced methods including VideoChat-R1 and TimeZero. As shown in Table R2, Divid continues to outperform or match these state-of-the-art models under the same settings.
>
> **Error Analysis:**
> We have added a detailed visualization and analysis of failure cases in the supplementary material. As illustrated in Figure 4, we examine two representative examples from the ReXTime benchmark and compare Divid with the full-attention baseline. In both cases, although the models correctly answer the multiple-choice questions, their predicted temporal intervals receive low tIoU scores, demonstrating that producing the correct answer does not necessarily imply correct temporal grounding.
> Our analysis shows a consistent pattern: when the question-relevant region and the answer-relevant region do not overlap, the baseline tends to localize only the question cue, completely missing the answer interval. Divid mitigates this issue by predicting a longer segment that covers both the question cue and the answer-relevant region, leading to higher but still imperfect tIoU. We attribute this behavior to the characteristics of existing training data, where question and answer segments typically lie within the same temporal region, providing limited supervision for cross-interval reasoning such as “before’’ or “after’’ questions.
> In future work, we will expand cross-interval training data to improve the model’s cross-interval temporal grounding ability.
>
> ### Table R2. Performance Comparison on Charades-STA.
>
> | Method       | R@0.5 | R@0.7 | mIoU |
> |--------------|-------|-------|------|
> | VideoChat-R1 | 71.7  | 50.2  | 60.8 |
> | TimeZero     | 72.5  | 47.9  | -    |
> | Ours         | 72.6  | 50.8  | 61.3 |

---

> ### Author Response · Authors · 2025-11-25
>
> >[**Q2**]. The design of the dual-branch decoder and the token-level router is presented in a relatively engineering-driven fashion, with little attempt at formalizing why or under what conditions disentangled decoder branches + soft dynamic fusion should yield better temporal localization or reduce hallucination. There is no theoretical analysis of capacity, representation expressivity, or inductive bias gains. The discussion about why fusion at the token level (rather than head-level, block-level, or post-attention fusion) is beneficial is thin and left to ablations.
>
> [**A2**]. Thank you for the reviewer’s comments. We address the question in three parts as follows.
>
> **Why and under what conditions disentangled decoder branches + soft dynamic fusion improve temporal localization and reduce hallucination?**
>
> Existing Video-LLMs typically process dense temporal tokens and high-resolution spatial tokens within a single unified attention space, which often causes temporal cues to be overshadowed by visually salient but semantically irrelevant spatial patterns. Prior studies such as MASH-VLM highlight that such entanglement leads to temporal confusion and scene-driven hallucination, especially in long videos where temporal dependencies are global but spatial evidence is sparse. Our model resolves this issue by explicitly disentangling the reasoning pathways inside the decoder. The temporal branch first builds a stable representation of long-range motion structure and generates cross-modal attention scores that identify the frames most relevant to the query. These scores then guide the spatial branch to focus high-resolution reasoning strictly within the correct temporal neighborhood, avoiding the dilution of key evidence seen in uniform frame sampling, where many selected frames are unrelated to the query and thus weaken temporal grounding. The token-level soft router further adapts the fusion for each query token, allowing the model to emphasize the modality most aligned with the token’s semantics and preventing irrelevant visual scenes from dominating temporal inference.
> This architecture yields clear benefits under conditions of (1) long-range temporal dependencies, (2) sparse but crucial spatial evidence, (3) queries containing explicit temporal relations, (4) strong token-density imbalance between spatial and temporal tokens, and (5) the compute constraints typical in long-video scenarios. Under these conditions, temporally guided keyframe selection combined with disentangled, token-level fusion provides a strong inductive bias for precise temporal localization while substantially reducing hallucination.

---

> ### Author Response · Authors · 2025-11-25
>
> [**A2**].
>
> **The theoretical analysis of capacity, representation expressivity, or inductive bias gains.**
> A complete theoretical characterization of the representational capacity induced by disentangled decoder branches is indeed a valuable research direction, though it lies beyond the scope of the current work. To address the reviewer’s concern, we have strengthened our explanation by clarifying the architectural motivations from a theoretical perspective and by providing additional empirical evidence.
> From the perspective of capacity and expressivity, disentangling spatial and temporal modeling increases the decoder’s effective functional capacity by mitigating interference between dense temporal tokens and high-resolution spatial tokens. This separation enables each branch to specialize in a distinct class of functions. Specifically, the temporal branch focuses on global temporal dependency modeling, while the spatial branch focuses on fine-grained spatial reasoning. These roles jointly expand and better structure the model’s overall hypothesis space.
>
> Regarding inductive bias, the architecture introduces a temporal-first and spatial-refine computation path that aligns with the intrinsic structure of video understanding. The temporal branch encourages a global motion-scanning bias, and the spatial branch emphasizes detail-oriented evidence retrieval guided by temporal cues. In addition, the token-level soft router provides a query-adaptive inductive bias that allows each token to selectively emphasize temporal or spatial information depending on its semantics.
> To complement this conceptual explanation, we have added a detailed empirical analysis of the soft router’s behavior in the appendix (Figure 3). For every query token across decoder layers, we compute temporal and spatial routing weights, average them to obtain token-level fusion scores, and visualize the 100 tokens with the strongest temporal or spatial preference. Tokens with explicit temporal semantics such as “then,” “when,” and “start” consistently receive higher temporal routing weights, while tokens referring to spatial entities or fine-grained visual details such as “lady,” “dog,” “applying,” and “holding” show strong spatial preference. These findings demonstrate that the router learns meaningful, query-aligned fusion patterns rather than assigning weights arbitrarily, which provides concrete empirical evidence that the intended inductive biases emerge in practice.
>
> **Why fusion at the token level (rather than head-level, block-level, FFN-level, or output-level) is beneficial?**
>
> LLMs operate under causal attention, meaning that during inference each token is generated sequentially and retrieves evidence independently. Considering this property, it is natural to aggregate spatial and temporal cues at the granularity of each query token. Token-level fusion enables the model to dynamically decide which modality to emphasize for each generated token, aligning the fusion mechanism with the autoregressive decoding process inherent to LLMs.
> Our ablations in Table R3 further validate this design choice. Output-level fusion performs integration only once at the end of decoding, which is too late for stable temporal grounding. Block-level fusion interacts only every few layers, resulting in insufficient cross-modal exchange and degraded performance. Head-level fusion merges signals immediately after QKV projection, which disrupts head-specific feature distributions and harms attention modeling. FFN-level fusion delays spatial–temporal interaction until after attention, limiting its impact on token-dependent evidence retrieval. In contrast, our token-level fusion, applied immediately after attention and before the FFN at every layer, consistently yields the strongest performance across all metrics. This placement injects temporal guidance early and continuously while preserving the structural stability of attention, leading to more reliable and accurate temporal localization.
>
> ### Table R3. Ablation of fusion methods and placements.
> | Method                | Charades-STA R1@0.5 | Charades-STA mIoU | ReXTime mIoU | ReXTime Acc@IoU |
> |-----------------------|----------------------|--------------------|--------------|------------------|
> | Output-level          | 45.80                | 42.51              | 26.44        | 16.63            |
> | Block-level (4 layers)| 49.66                | 45.98              | 28.23        | 17.24            |
> | Block-level (2 layers)| 50.37                | 47.04              | 29.63        | 18.20            |
> | Head-fusion           | 50.41                | 46.95              | 29.34        | 18.21            |
> | FFN-fusion            | 50.77                | 47.12              | 29.58        | 18.30            |
> | Ours           | 51.37                | 47.33              | 29.71        | 18.57            |

---

> ### Author Response · Authors · 2025-11-25
>
> >[**Q3**]. The TempGCap construction process involves multiple automated and manual annotation stages (as in Figure 2), but more detail should be provided about inter-annotator agreement, filtering criteria, and possible biases introduced (e.g., via pseudo-untrimmed synthesis). Even though Table 13 breaks out annotation strategies, the risk of subtle domain leakage or noise propagation is underexplored.
>
> [**A3**]. Thank you for the reviewer’s comments. We address the question in three parts as follows.
>
> **More Detail of Dataset Construction:**
> We have added a substantially expanded description of our dataset construction pipeline in the Appendix, covering inter-annotator consistency, filtering criteria, and potential sources of bias. Specifically, Sub1 incorporates datasets with existing manually annotated temporal boundaries such as DiDeMo and ActivityNet Captions, and for HACS we generate segment-level captions using Tarsier and apply a two-stage filtering pipeline that includes rule-based removal of overly long or shot-change–dominated captions, static-content scoring with Qwen2-7B, and random human audits to verify alignment and descriptive accuracy. To ensure consistency with the detailed captions produced by Tarsier, we adopt instruction formats such as “Describe … in detail” in all training samples.
>
> For Sub2, where untrimmed videos are recovered, we align each captioned clip to its original timestamps, keep the caption unchanged, and treat surrounding context only as distractors. We further refine segment boundaries using ViT-based similarity between boundary frames and their neighbors, expanding timestamps when similarity falls below 0.9, capping the adjustment range, and discarding segments with unstable transitions. Additional quality-control steps remove corrupted downloads, abnormally short videos, and duplicates, followed by random human audits to ensure that refined segments remain semantically correct and that added context does not introduce misleading artifacts.
>
> For Sub3, where pseudo-untrimmed sequences are synthesized, we concatenate 2–4 additional video clips in random order, place the target clip at varying positions, and recompute timestamps accordingly. We apply textual filtering using BERT similarity and visual filtering using UMT features to remove distractors that are too similar to the target, and we also filter repeated captions. Manual spot-checking further ensures that synthesized contexts are reasonable and do not unintentionally overlap semantically with the target clip. Since each target clip already carries a high-quality and semantically accurate caption, and our task only requires describing the content within the specified temporal interval, the intrinsic alignment between the segment and its caption remains precise. The pseudo-untrimmed sequence serves merely as surrounding context, and we uniformly vary the target clip’s position to avoid structural bias. Combined with the preservation of original segment boundaries and the applied textual and visual dissimilarity filtering, this synthesis process does not introduce significant bias and does not affect temporal grounding accuracy.

---

> ### Author Response · Authors · 2025-11-25
>
> [**A3**].
>
> **Domain Leakage:**
> We have added a detailed description of our data leakage prevention measures in the Appendix. Specifically, we have implemented a strict three-stage filtering strategy to ensure that TempGCap does not overlap with any downstream evaluation benchmarks. First, we remove any video whose name-based identifier (such as youtube_id or other dataset-specific video IDs) matches those appearing in our evaluation datasets, guaranteeing that no training sample contains content from test videos. Second, to further prevent domain-level leakage, we manually exclude video sources whose content or annotation style closely resembles the target benchmarks. In particular, we filter out Charades, Charades-Ego, Ego4D, and TGIF, which naturally overlap with temporal grounding and VideoQA domains. Third, for datasets that provide official training, validation, and test splits, including DiDeMo, HACS, ActivityNet Captions, and Kinetics-700, we retain only the training split and explicitly exclude all validation and test videos to avoid any accidental overlap. This comprehensive filtering procedure ensures that TempGCap remains clean, non-overlapping, and free of both data-level and domain-level leakage under our zero-shot evaluation protocol.
>
> **Noise Propagation:**
> In TempGCap, each construction stage includes explicit filtering to prevent the accumulation of noise. In Sub1, captions generated by Tarsier are filtered through rule-based checks, static-content scoring with Qwen2-7B, and random human audits. In Sub2, we refine temporal boundaries using ViT-based similarity and discard segments with ambiguous transitions, while also filtering corrupted or abnormal videos. In Sub3, pseudo-untrimmed sequences are synthesized only after textual and visual dissimilarity filtering, followed by manual spot-checking.
> Although automated steps exist, every target clip retains a high-quality caption, and our task only requires describing the specified temporal interval, which stabilizes the supervision even when surrounding context is synthesized. Uniformly varying the target clip’s position and preserving its original boundaries further prevent structural biases.
> Empirically, TempGCap’s quality is validated in Table 8: despite being far smaller than Momentor-10M, it yields higher performance on Charades-STA and ReXTime, indicating that our hybrid filtering pipeline effectively controls noise and provides more reliable temporal supervision.

---

> ### Author Response · Authors · 2025-11-25
>
> >[**Q4**]. Minor Issues in Exposition:
> There are a few places where notation is not adequately defined or is overloaded (see Section 3.2, Page 4-5).
> Typographical issues, e.g., "the the selected spatial features" (Page 5).
> Some results tables (Table 1, Table 2) reproduce highly dense comparisons, but occasionally lack scale clarity (e.g., 1.5B vs 7B), and the presentation would benefit from clearer highlighting to aid comparison.
>
> [**A4**].
> We thank the reviewer for the helpful suggestions. We have carefully revised Section 3.2 to ensure that every variable is clearly introduced before use. We also corrected all typographical errors, and additionally performed a full proofreading pass to eliminate similar issues throughout the paper. For Table 1 and Table 2, we improved the presentation by using light blue to highlight models smaller than 7B and light yellow to highlight models larger than 10B, making scale comparisons clearer at a glance.
>
> >[**Q5**]. Can the authors provide an explicit composite training loss equation, including all task-specific loss terms and weighting, as actually used in implementation (e.g., for temporal localization vs. QA)? How are thresholds, negative sampling, and class imbalance handled during optimization?
>
> [**A5**]. Thank you for the question. We have added an explicit composite training loss equation in the Appendix. All tasks in our framework, including timestamp-guided captioning, temporal grounding, and grounded VideoQA, are trained with a single unified autoregressive language modeling loss. We do not introduce task-specific losses, weighting terms, or auxiliary objectives. Temporal boundaries are represented directly as text tokens, so no regression heads, thresholds, or margin-based criteria are used. We also do not apply explicit negative sampling or class-balancing strategies during optimization.
>
>
> >[**Q6**]. What is the rationale for choosing a token-level softmax on projected text features for branch fusion? Did the authors experiment with deeper gating modules, or alternative fusion placements (e.g., block, head, or output level)?
>
> [**A6**]. Thank you for the question. We have added a detailed explanation and the corresponding ablation results in Table R4. Our choice of a token-level softmax on projected text features is motivated by the autoregressive nature of LLM decoding: each token retrieves evidence independently, so allowing fusion at the same token granularity enables the model to dynamically adjust how much temporal or spatial information it uses at every decoding step. The gating module is intentionally lightweight to maintain stability and avoid additional optimization complexity.
>
> We also experimented with deeper gating modules as well as multiple fusion placements. Table R4 shows that output-level fusion performs the weakest because spatial–temporal interaction happens only once at the end. Block-level fusion reduces interaction frequency and degrades temporal grounding. Head-level fusion disrupts attention head structure, while FFN-level fusion introduces fusion too late in the computation. Our method, which performs token-level fusion immediately after attention and before the FFN in every layer, achieves the best performance on both Charades-STA and ReXTime. These results confirm that token-level fusion is the most effective and stable design among all tested alternatives.
>
> ### Table R4. Ablation of fusion methods and placements.
> | Method                | Charades-STA R1@0.5 | Charades-STA mIoU | ReXTime mIoU | ReXTime Acc@IoU |
> |-----------------------|----------------------|--------------------|--------------|------------------|
> | Output-level          | 45.80                | 42.51              | 26.44        | 16.63            |
> | Block-level (4 layers)| 49.66                | 45.98              | 28.23        | 17.24            |
> | Block-level (2 layers)| 50.37                | 47.04              | 29.63        | 18.20            |
> | Head-fusion           | 50.41                | 46.95              | 29.34        | 18.21            |
> | FFN-fusion            | 50.77                | 47.12              | 29.58        | 18.30            |
> | Ours           | 51.37                | 47.33              | 29.71        | 18.57            |

---

> ### Author Response · Authors · 2025-11-25
>
> >[**Q7**]. Do the temporal and spatial branches demonstrably specialize in the types of reasoning intended (e.g., is the spatial branch consistently responsible for frame-level details, or do branches collapse under certain queries)? Would the authors consider adding qualitative analyses or illustrative attention maps?
>
> [**A7**]. Thank you for the insightful question. Our architecture is explicitly designed to enforce functional specialization between the temporal and spatial branches, and our new analyses confirm that they indeed operate in the intended manner. The temporal branch processes densely sampled but low-resolution frames, allowing it to capture long-range motion dynamics and accurately determine which portions of the video are relevant to the query. In contrast, the spatial branch receives only a small set of high-resolution keyframes, enabling it to focus its capacity on fine-grained details such as objects, textures, and subtle action cues. Since temporal attention directly guides the Top-K keyframe selection, the spatial branch is intrinsically conditioned on temporally identified regions of interest. These differences in input density, spatial resolution, and routing mechanism naturally prevent the two branches from collapsing and encourage them to specialize.
>
> To verify this behavior, we include qualitative analyses of key-frame selection in the appendix (Figure 5). Across all decoder layers, most selected frames concentrate within the ground-truth temporal window, demonstrating that the temporal branch reliably highlights the correct video region. A small fraction of selected frames falls just outside the annotated segment; however, these frames typically contain semantically related activities, explaining why they still attract temporal attention. Overall, the selection patterns show that the temporal branch effectively localizes relevant moments, while the spatial branch focuses on detailed visual understanding within those moments.
>
> We further analyze the interpretability of our soft router by examining token-wise routing weights (Figure 3). Tokens with temporal semantics (e.g., “when,” “then,” “start”) consistently receive higher temporal weights, while tokens describing spatial entities or fine-grained visual attributes (e.g., “lady,” “dog,” “holding,” “applying”) exhibit stronger spatial preferences. This demonstrates that the router does not allocate fusion weights arbitrarily: it learns semantically meaningful, query-dependent strategies aligned with disentangled spatial–temporal reasoning.
>
> Together, the architectural constraints, key-frame selection patterns, and semantically meaningful routing weights consistently indicate that the spatial and temporal branches specialize as intended, rather than collapsing into redundant behavior.

---

> ### Author Response · Authors · 2025-11-25
>
> >[**Q8**]. Given the mixture of manual, pseudo, and synthesized captions in TempGCap, how is data leakage prevented on evaluation sets, and have any systematic biases been observed during annotation or curation phases?
>
> [**A8**]. Thank you for the question. We have provided a consolidated clarification addressing both data-leakage prevention and systematic bias mitigation in the construction of TempGCap, with expanded details included in the Appendix. To prevent data leakage, we apply a strict multi-stage filtering strategy. We first remove all videos whose identifiers, such as YouTube IDs or dataset-specific video IDs, appear in any evaluation sets, ensuring that no training instance contains content from test videos. We then further guard against domain-level leakage by excluding datasets whose content style or annotation format naturally resembles downstream temporal grounding benchmarks; specifically, we filter out Charades, Charades-Ego, Ego4D, and TGIF to avoid introducing domain priors that could inflate zero-shot performance. Finally, for datasets with official train/validation/test splits, including DiDeMo, HACS, ActivityNet Captions, and Kinetics-700, we retain only the official training split and discard all validation and test videos. Together, these measures ensure that TempGCap remains entirely non-overlapping with evaluation benchmarks and free of domain-level leakage.
>
> To address potential systematic biases arising from the mixture of manual, pseudo-untrimmed, and synthesized captions, we design dedicated quality-control procedures for each construction stage. In Sub1, where manually annotated temporal boundaries are combined with caption generation from Tarsier, we apply a two-stage filtering pipeline consisting of rule-based removal of low-quality captions, static-content scoring with Qwen2-7B, and random human audits to control noise and ensure semantic accuracy. In Sub2, where untrimmed videos are recovered, we refine segment boundaries via ViT-based inter-frame similarity, adjust timestamps when boundary transitions appear unstable, and remove corrupted or abnormal videos. Human spot-checking verifies that reintroduced context does not produce misleading cues or boundary drift. In Sub3, where pseudo-untrimmed sequences are synthesized, we mitigate potential structural bias by randomizing the target clip’s position in the sequence and enforcing textual and visual dissimilarity filtering using BERT and UMT features to prevent distractors that are overly similar to the target. Each target clip retains its original high-quality caption, meaning the core alignment between segment and description remains intact. Additional manual spot-checking ensures that surrounding synthesized context does not inadvertently overlap semantically with the target.
>
> Across all stages, these filtering and verification mechanisms prevent noise accumulation and reduce the risk of systematic annotation or curation bias. Empirical results in Table 8 further validate data quality: despite being considerably smaller than Momentor-10M, TempGCap leads to higher temporal grounding performance on Charades-STA and ReXTime, demonstrating that our hybrid filtering pipeline effectively controls noise while preserving high temporal fidelity.

---

> ### Author Response · Authors · 2025-11-25
>
> >[**Q9**]. The most important is, I noticed that the comparative methods adopted by the authors lag behind the latest state-of-the-art research. Recently proposed models such as Qwen2.5-VL, VideoChat-R1, and TimeZero (introduced about six months ago) have achieved strong performance across various temporal localization and general video question answering benchmarks. What are the advantages of the authors’ proposed method compared with these approaches?
>
> [**A9**]. Thank you for the reviewer’s constructive comment. We agree that incorporating recent state-of-the-art models is essential for a fair and comprehensive comparison. We address this concern from two perspectives: performance relative to the Qwen2.5-VL series and extended comparisons with VideoChat-R1 and TimeZero.
>
> First, Table 1 in the main paper already includes comparisons with the Qwen2.5-VL family. Under the zero-shot setting, our 1.5B model surpasses Qwen2.5-VL 7B in mIoU on Charades-STA, and our 7B model achieves performance comparable to Qwen2.5-VL 72B, despite using a significantly smaller backbone.
>
> To further address the reviewer’s request for comparisons with the most recent advanced models, we additionally evaluate against VideoChat-R1 and TimeZero, both of which are fine-tuned on the Charades-STA training set. For fairness, we also fine-tune our 7B model on the same training split, and the results are presented in Table R5. Notably, both competing methods employ GRPO-based reinforcement learning with task-specific temporal reward functions and rely on the stronger Qwen2.5-VL backbone. In contrast, our model builds on the earlier Qwen2 series and does not use reinforcement learning. Despite these inherent advantages for competing approaches, our 7B model achieves the best overall performance on Charades-STA.
>
> These findings highlight the effectiveness of our disentangled architecture and the strong temporal grounding capability gained through TempGCap. Overall, our method remains competitive with—and in several cases surpasses—the latest state-of-the-art models, even under comparatively restrained backbone capacity and training strategies.
>
> #### Table R5. Performance Comparison on Charades-STA.
>
> | Method       | R@0.5 | R@0.7 | mIoU |
> |--------------|-------|-------|------|
> | VideoChat-R1 | 71.7  | 50.2  | 60.8 |
> | TimeZero     | 72.5  | 47.9  | -    |
> | Ours         | 72.6  | 50.8  | 61.3 |

---

> ### Author Response · Authors · 2025-11-26
>
> Thank you again for your careful and constructive review. For clarity, we summarize below the key improvements made in response to your comments (Q1–Q9), grouped into a few concise themes:
>
> * **Expanded Experimental Coverage & More Comprehensive Comparisons (Q1, Q9)**
>   We added comprehensive evaluations against the latest SOTA Video LLMs—including **VideoChat-R1**, **TimeZero**, and **Qwen2.5-VL**—and enriched the generalization analysis across both grounding-focused benchmarks and standard VideoQA datasets. Additional failure-case analyses provide deeper insight into model behavior.
>
> * **Clearer Architectural Motivation & New Empirical Evidence (Q2, Q6, Q7)**
>   We strengthened the conceptual justification for our **disentangled dual-branch decoder**, detailed the inductive biases it introduces, and supported it with new analyses of **token-level routing patterns**, **branch specialization**, and extensive ablations across fusion strategies and placements.
>
> * **Substantially Enriched Dataset Construction Details (Q3, Q8)**
>   We added rigorous descriptions of TempGCap’s pipeline, including **inter-annotator consistency**, **boundary refinement**, **multi-stage filtering**, **bias mitigation**, and **strict data-leakage prevention**. These clarifications show why TempGCap provides higher-fidelity temporal supervision than prior large-scale datasets.
>
> * **Improved Clarity, Notation, and Training Explanation (Q4, Q5)**
>   We corrected notation, refined exposition, improved table readability, and provided the full composite training objective, confirming that all tasks are trained using a unified autoregressive loss without auxiliary heads or thresholds.
>
> Together, these revisions significantly **strengthen the technical clarity, empirical rigor, and overall contribution** of our work. If you find the updated analysis and results satisfactory, we would be sincerely grateful if you could consider **raising your initial score**. We are happy to address any additional questions during the discussion phase.
>
> Thank you again for your valuable time and thoughtful feedback.

---

> ### Comment · Reviewer_KBdS · 2025-11-26
>
> I thank the authors for their detailed response. Although the proposed architecture has certain limitations (e.g., improving Temporal Grounding may trade off with General QA accuracy), I do not view this as a fatal issue when the model is positioned as a specialized MLLM for temporal localization. I acknowledge that matching SFT-based methods purely with RL is indeed challenging, and the proposed dataset seems to be a valuable contribution that could enable future research in this direction.
>
> Consequently, I am willing to raise my score to a positive rating. I hope the authors could confirm their commitment to fully open-sourcing both the code and the training data in the future.

---

> > ### Author Response · Authors · 2025-11-27
> >
> > Thank you for your positive reassessment. We will fully open-source the code and the TempGCap dataset in the future. We sincerely appreciate your constructive feedback.

---

### Official Review · Reviewer_1ujf · 2025-10-31

**Soundness:** 3
**Presentation:** 3
**Contribution:** 3
**Rating:** 6
**Confidence:** 4

**Summary:**

This paper presents DIVID, a novel dual-branch framework designed to disentangle spatial and temporal modeling within the LLM decoder for temporally grounded video understanding, with key innovations including: (1) a temporal branch that processes low-resolution, densely sampled frames to capture long-range dynamics; (2) a spatial branch that selects high-resolution keyframes via temporal attention for fine-grained visual reasoning; and (3) a token-level spatio-temporal soft-router that adaptively fuses features conditioned on the input query. To support training, the authors introduce TempGCap, a large-scale instruction-tuning dataset containing 559K timestamp-grounded video-text pairs with high temporal precision and diverse video coverage. Experiments on Charades-STA, CG-Bench, NExT-GQA, and ReXTime benchmarks demonstrate that DIVID achieves state-of-the-art performance in both temporal grounding and grounded video QA tasks while significantly reducing computational costs, establishing a new efficiency-accuracy trade-off for video understanding architectures.

**Strengths:**

1. The idea of disentangling spatial and temporal modeling inside the LLM decoder is novel and well-motivated. The soft-router mechanism is lightweight yet effective, enabling adaptive fusion at the token level.
2. DIVID outperforms many strong baselines, including larger models (e.g., 72B Qwen2-VL), especially in temporal grounding tasks. The 1.5B model already surpasses several 7B models, showing excellent parameter efficiency.
3. TempGCap is a valuable contribution to the community. It combines manual and automatic annotations, covers diverse video domains, and provides fine-grained temporal supervision, filling a gap in existing datasets.
4. The paper includes extensive ablations, efficiency analysis, and generalization studies. The authors carefully analyze the impact of each component (e.g., keyframe selection, soft-router, dataset quality), which strengthens the credibility of the results.

**Weaknesses:**

1. While the disentanglement is well-executed, the contributions are mostly at the module level, lacking a more fundamental architectural breakthrough.
2. The soft-router is effective but lacks interpretability. There is no analysis of how the gating weights vary across query types or whether the router learns semantically meaningful fusion strategies.

**Questions:**

1. DIVID performs well on grounding tasks but shows limited gains on general VideoQA (e.g., MSRVTT-QA). What do you think limits its generalization to non-grounding tasks?
2. Should the visual encoder (ViT-G/14) be fine-tuned during training to improve fine-grained temporal grounding?

---

> ### Author Response · Authors · 2025-11-25
>
> We sincerely appreciate your constructive feedback, positive evaluation of our contributions, and recognition of the novelty and effectiveness of our Divid framework. Your comments regarding both the strengths and remaining concerns have been highly valuable for improving our paper. We have carefully addressed all your questions and suggestions in the detailed response below and have incorporated all corresponding revisions and additional experimental results in the updated manuscript (highlighted in blue). We genuinely hope that our detailed explanations and the additionally supplemented results with our best efforts will give us the precious opportunity to raise the evaluation score of our work in your perspective.
>
> **Our Responses to Paper Weaknesses:**
> >[**Q1**]. While the disentanglement is well-executed, the contributions are mostly at the module level, lacking a more fundamental architectural breakthrectly reshape how the LLM decoder processes video information, introducing architectural behaviors that are absent in existinrough.
>
> [**A1**]. Thanks for your comment. We agree that Divid adopts the common visual-encoder → projector → LLM pipeline used in many VideoLLMs. However, our core contributions go beyond module-level additions because our Divid framework directly reshapes how the LLM decoder processes spatio-temporal information.
>
> First, Divid performs spatio-temporal disentanglement inside every decoder layer rather than only at the input stage. Prior attempts extract temporal or spatial tokens externally and then feed them into an unchanged LLM decoder in which all tokens still interact within a unified attention space. In contrast, Divid restructures each decoder block into two parallel branches, the Temporal Perception branch and the Spatial Perception branch, so that temporal and spatial information remain explicitly disentangled throughout all decoding layers. This changes the internal attention factorization pattern of the decoder and therefore constitutes a decoder-level architectural modification rather than an external module design.
>
> Second, our dynamic spatio-temporal soft router introduces a new control-flow mechanism inside the LLM. The router computes token-wise, query-conditioned routing weights that determine how each text token integrates temporal and spatial cues at every layer. This design alters the topology of information flow in a way conceptually similar to lightweight Mixture-of-Experts routing. It is not a static weighting scheme but a token-specific and layer-specific process that fundamentally changes how the decoder incorporates video information.
>
> Third, our query-guided keyframe selection forms a temporal-to-spatial dependency loop that is more than a preprocessing step. The temporal branch first produces text-conditioned saliency scores, and these scores then determine the high-resolution keyframes used by the spatial branch. This establishes a feedback structure between the two branches, creating a new form of query-aware execution flow that is absent in VideoLLMs relying on uniform or heuristic sampling.
>
>
> >[**Q2**]. The soft-router is effective but lacks interpretability. There is no analysis of how the gating weights vary across query types or whether the router learns semantically meaningful fusion strategies.
>
> [**A2**]. Thank you for the constructive suggestion. We have added a detailed analysis of how the soft router’s gating weights vary across different query tokens, and the corresponding visualizations are now included in the appendix (Figure 3). To obtain these results, we compute the temporal and spatial routing weights of every query token across all decoder layers, average them to derive token-wise fusion scores, and then rank all tokens based on their temporal or spatial preference. We visualize the top 100 tokens for each branch.
> As shown in Figure 3(a), tokens with clear temporal semantics—such as “then,” “when,” and “start”—consistently receive higher temporal routing weights. In contrast, Figure 3(b) shows that tokens referring to spatial entities or fine-grained visual details, including nouns such as “lady” and “dog” and action verbs such as “applying” and “holding,” tend to receive higher spatial routing weights.
> These findings indicate that the router does not allocate weights arbitrarily. Instead, it learns semantically meaningful and query-dependent fusion behaviors that align with the goal of disentangled spatial–temporal reasoning. This analysis provides clear evidence of the interpretability of our soft router and further supports the effectiveness of incorporating token-level adaptive routing into the overall architecture.

---

> ### Author Response · Authors · 2025-11-25
>
> >[**Q3**]. Divid performs well on grounding tasks but shows limited gains on general VideoQA (e.g., MSRVTT-QA). What do you think limits its generalization to non-grounding tasks?
>
> [**A3**]. Thank you for the thoughtful question. The relatively small gains of Divid on general VideoQA benchmarks such as MSRVTT-QA are expected, given our architectural focus and training objectives.
> First, Divid is intentionally designed for temporally grounded video understanding rather than short-video content recognition. Our dual-branch decoder, query-guided keyframe selection, and token-level spatio-temporal routing explicitly target long-range temporal localization and reasoning. In contrast, benchmarks such as MSRVTT-QA, MSVD-QA, and ActivityNet-QA primarily evaluate object presence, scene attributes, and short-range actions, with minimal emphasis on multi-second temporal dependencies. As a result, the strengths of our disentangled temporal modeling do not directly translate into large gains on these appearance-centric benchmarks. Similar trends have been observed in other grounding-focused VideoLLMs such as LLaVA-ST [1] and TOGA [2], which likewise report limited improvements on these datasets.
> Second, our instruction-tuning dataset TempGCap is specifically constructed to provide timestamp-grounded supervision, enhancing the model’s ability to understand when events occur and how temporal structures unfold over long videos. While this supervision greatly benefits temporal grounding tasks, it is not the primary focus of generic VideoQA benchmarks, which helps explain the naturally smaller improvements observed there.
> Despite not being optimized for these tasks, Divid still achieves competitive performance compared with recent VideoLLMs, demonstrating that our disentangled modeling does not hinder generalization. Our goal is to advance temporally grounded long-video reasoning, where Divid delivers substantial and consistent improvements while maintaining solid performance on general VideoQA.
>
> [1] LLaVA-ST: A multimodal large language model for fine-grained spatial-temporal understanding.
>
> [2] TOGA: Temporally Grounded Open-Ended Video QA with Weak Supervision.
>
> >[**Q4**]. Should the visual encoder (ViT-g/14) be fine-tuned during training to improve fine-grained temporal grounding?
>
> [**A4**]. Thank you for the reviewer’s question. We have extended our analysis by adding a new ablation study on visual encoder fine-tuning. As shown in Table R1, fine-tuning the visual encoder indeed brings consistent improvements in fine-grained temporal grounding.
> Since ViT-g/14 contains approximately 1B parameters, fully fine-tuning it would incur substantial computational overhead. To make the ablation more tractable, we conducted the fine-tuning experiments using SigLIP (~400M parameters), which is considerably more efficient. Although ViT-g/14 is larger, it is pretrained at a lower resolution (224×224), whereas SigLIP is pretrained at 384×384. We observe that when both encoders are frozen, they achieve highly similar performance across all metrics, indicating that SigLIP serves as a reliable surrogate for evaluating the effect of encoder fine-tuning.
> When we fine-tune SigLIP during Stage II training, we observe consistent improvements on both Charades-STA and ReXTime. These results demonstrate that visual encoder fine-tuning further enhances temporal grounding performance and is therefore a beneficial choice when computational resources permit.
>
> ### Table R1. Ablation of Visual Encoder
> | Visual Encoder | Charades R1@0.5  | Charades mIoU  | ReXTime mIoU  | ReXTime Acc@IoU  |
> |---------|------------------|----------------|---------------|------------------|
> | ViT-g   | 51.37            | 47.33          | 29.71         | 18.57            |
> | SigLIP  | 51.08            | 47.09          | 29.45         | 18.32            |
> | SigLIP (FT)  | 52.16            | 47.77          | 29.98        | 18.79            |

---

> ### Comment · Reviewer_1ujf · 2025-11-27
>
> Thank you for the thorough and insightful responses. I’m satisfied with the added analyses, which strengthen the paper’s validity. I maintain my original rating.

---

> > ### Author Response · Authors · 2025-11-27
> >
> > Thank you for your thorough review and positive feedback. We appreciate your time and constructive assessment, and we are glad that the additional analyses addressed your concerns. Thank you again for your thoughtful evaluation.

---

### Author Response · Authors · 2025-11-30

We sincerely thank the ACs and reviewers for their time and constructive feedback. We have carefully **addressed all concerns** and **strengthened the paper** with additional ablations, qualitative analyses, and clarifications (the manuscript has expanded **from 20 to 27 pages**). We appreciate that **all reviewers expressed satisfaction** with our responses, and that **the only initially negative reviewer has updated their score to a positive rating (6)**, with all reviewers having set their scores to **6, 6, and 6** as of November 26.
Please refer to our detailed comments for more information. We truly appreciate everyone’s insightful feedback once again.

---

### Author Response · Authors · 2025-12-03

Dear ACs,

Thank you for your time and effort. We are providing this brief summary to facilitate your assessment of our paper. We understand the difficult circumstances and deeply appreciate your involvement in this process. We hope that our summary can be helpful for you to make a decision on our submission.

First, regarding the architectural motivation, as clarified in the rebuttal, our work is driven by the observation that existing Video LLMs entangle spatial and temporal information within the decoder, leading to temporal confusion and inefficiency on long videos. Divid directly addresses this limitation through a dual-branch framework that explicitly disentangles spatial and temporal modeling within the LLM decoder: the Temporal Branch processes densely sampled, low-resolution frames to effectively capture long-range video dynamics, the Spatial Branch attends to high-resolution keyframes selected via temporally guided attention, and a spatio-temporal soft-router adaptively fuses temporal and spatial cues at the token level, enabling more effective and efficient temporal understanding.

Second, regarding empirical validation and interpretability, we strengthened our analysis with comprehensive ablations on fusion strategies, encoder configurations, and architectural variants, along with newly added comparisons against the latest state-of-the-art Video LLMs, including VideoChat-R1 and TimeZero. In addition, qualitative visualizations of routing weights and cross-layer keyframe selection demonstrate that the branches specialize as intended and that the adaptive fusion aligns with the semantics of query tokens, and we also include an analysis of representative failure cases.

Finally, we addressed the remaining reviewer concerns through: 1) a fully expanded and rigorously documented TempGCap construction pipeline, including annotation workflows, multi-stage filtering, boundary refinement, and strict data-leakage prevention; 2) additional clarifications on the unified autoregressive training objective, notation refinements, and improved table readability; and 3) implementation optimizations that reduce inference latency and enhance the practical efficiency of the system.

During the rebuttal phase, all reviewers expressed satisfaction with our responses. In particular, **the only initially negative reviewer updated his/her score to a positive rating of 6 on November 26, one day before the OpenReview incident on November 27**. Before the discussion period ended early, all reviewers had confirmed their final scores to be **6, 6, and 6**.

We solemnly state that we did not and would never attempt to use the leaked information to harass any community member or manipulate the review process in any way.



Sincerely yours,

Authors of paper #13461

---

### Meta-Review · Area_Chair_xHsW · 2026-01-09

**Summary:**

The paper introduces DIVID, a dual-branch decoder architecture that disentangles spatial and temporal modeling, and TempGCap, a large-scale temporal grounding dataset. Reviewers praise the parameter efficiency and the dataset's value for temporal grounding. Concerns persist regarding the lack of comparisons with the latest SOTA, the interpretability of the soft-router, and the limited generalization to non-grounding general VideoQA tasks.

**Reviewer Concerns:**

The author's rebuttal addressed most of the issues pointed out by the reviewers, especially providing key clarifications for the questions raised by Reviewer KBdS.

**Reviewer Scores:**

Reviewer KBdS is willing to raise the rating.

---

### Decision · Program_Chairs · 2026-01-26

Accept (Poster)